# DDIM Inversion as a Perturbation Amplifier: Breaking Mimicry Protection via Reconstruction Error Minimization

**Huming Qiu** [1]  **Peiyi Chen** [1]  **Mi Zhang** [1]  **Geng Hong** [1]  **Xiaoyu You** [2]  **Mi Wen** [3]  **Min Yang** [1]

## Abstract

Personalization techniques for image generation models have increasingly been misused for malicious purposes, including unauthorized style imitation and copyrighted content replication. In response, recent mimicry protection methods embed carefully designed perturbations into images to disrupt a model's ability to learn genuine semantic representations. Despite their growing adoption, the robustness of these protection mechanisms remains poorly understood, raising concerns about their reliability in real-world deployment. In this work, we present the first systematic analysis showing that DDIM inversion inherently acts as a perturbation amplifier, causing protected images to suffer severe structural distortions during reconstruction. Building on this observation, we propose DDIM Inversion-based Reconstruction Purification (DIRP), a novel purification approach that removes protective perturbations by explicitly minimizing DDIM inversion reconstruction error under perceptual constraints. Extensive experiments on six existing mimicry protection methods demonstrate that DIRP consistently outperforms seven state-of-the-art attack baselines, achieving superior perturbation removal while better preserving image quality. Our results expose fundamental vulnerabilities in current mimicry protection strategies and highlight the urgent need for more robust and principled defenses.

## 1. Introduction

In recent years, text-to-image (T2I) models (Rombach et al., 2022) have made remarkable progress in personalization, enabling the learning and generation of new styles or specific entities from as few as four example images (Ruiz et al., 2023). While this few-shot fine-tuning paradigm substantially reduces the dependence on large-scale personalization data, it also introduces significant risks of malicious misuse. Numerous real-world cases have shown that such models can be exploited to imitate the artistic styles of specific creators without authorization (Chen, 2023) or to generate pornographic content of targeted individuals (The Star, 2025), leading to serious copyright and privacy violations.

To mitigate this risk, mimicry protection methods achieve protection by injecting carefully designed protective perturbations into images before they are used for model personalization (Shan et al., 2023), have been downloaded millions of times (The Glaze Project, 2025). These perturbations aim to disrupt the learning process of diffusion models, making it difficult for them to capture genuine semantics (Truong et al., 2025). As a result, personalization either suffers from severe generation quality degradation or forces the model to encode detectable watermark. For example, Anti-DreamBooth (Van Le et al., 2023) optimizes perturbations that maximize the fine-tuning loss, thereby hindering the model's ability to learn the semantic structure of protected images, while DIAGNOSIS (Wang et al., 2023) introduces subtle geometric distortions through radial transformations, causing the generated outputs to carry detectable patterns. Since potential attackers may become aware of these protective perturbations and attempt to remove them, mimicry protection methods must exhibit sufficient robustness to withstand such removal attacks (Zhang et al., 2025).

However, despite demonstrating certain protective effects under standard personalization pipelines, the robustness of existing methods has not been systematically validated or thoroughly analyzed (Foerster et al., 2025). This lack of rigorous evaluation may lead to an overestimation of their reliability in real-world deployment, resulting in a false sense of security (Ye et al., 2025). Under the manifold hypothesis (Farghly et al.), T2I models tend to generate and model images that lie on the semantic manifold learned from

[1]Fudan University, Shanghai, China [2]East China University of Science and Technology, Shanghai, China [3]Shanghai University of Electric Power, Shanghai, China. Correspondence to: Mi Zhang <mi_zhang@fudan.edu.cn>, Min Yang <m_yang@fudan.edu.cn>.

*Proceedings of the 43rd International Conference on Machine Learning*, Seoul, South Korea. PMLR 306, 2026. Copyright 2026 by the author(s).

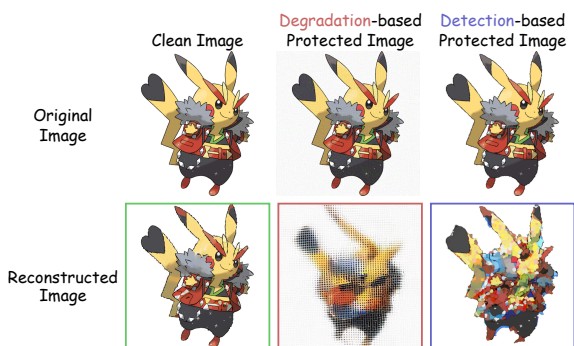

**Figure 1.** Examples of images reconstructed via DDIM Inversion. The clean image is faithfully reconstructed, while the protected image shows significant distortion. See Figure 5 for more examples.

the training data. Accordingly, we argue that the anomalous behaviors induced by existing mimicry protection methods (e.g., degradation in generation quality) may be due to the injected perturbations push images off the learned manifold, resulting in structured feature deformation (Martin et al., 2025). Such off-manifold deviations inherently limit the stealthiness and robustness of current protection strategies. This observation naturally raises the following question:

*Is there a method that can amplify and remove these protective disturbances?*

**Our Work.** In this paper, we provide an affirmative answer to this question by uncovering a previously overlooked but critical property of DDIM inversion (Song et al., 2020). We show that DDIM inversion exhibits a pronounced *amplification effect* on protective perturbations. As illustrated in Figure 1, even perturbations that are nearly imperceptible in the pixel space can induce disproportionately large reconstruction errors during DDIM inversion. We further present a theoretical analysis of this phenomenon and demonstrate that, compared to reconstruction paradigms such as VAE reconstruction or diffusion reconstruction, DDIM inversion follows a deterministic inversion denoising trajectory constrained by the learned semantic manifold (see section 3 for details). Along this trajectory, reconstruction errors are primarily governed by the Jacobian of the noise prediction network, rendering DDIM inversion particularly sensitive to protective perturbations.

Building on this insight, we propose a purification method for protective perturbations, termed DDIM Inversion-based Reconstruction Purification (DIRP). DIRP explicitly minimizes the reconstruction error of DDIM inversion under perceptual constraints, iteratively optimizing a purification residual that pulls protected images back toward the manifold learned by the diffusion model, thus ensuring effective learning for personalized methods. Our method enjoys several appealing properties: *(i) universality*, as it applies to

both degradation-based and detection-based mimicry protection methods; *(ii) black-box*, requiring no access to or knowledge of the underlying protection algorithms; *(iii) data-free*, relying solely on the protected images themselves; and *(iv) transferability*, generalizing across multiple T2I models.

In summary, our main contributions are as follows.

- We identify and analyze the *perturbation amplification effect* of DDIM inversion, revealing its ability to magnify imperceptible protective perturbations during reconstruction.

- Leveraging this property, we propose DIRP, a DDIM inversion-based reconstruction purification method that effectively removes protective perturbations introduced by mimicry protection methods by minimizing reconstruction error under perceptual constraints.

- We conduct a comprehensive evaluation on six state-of-the-art mimicry protection methods. Experimental results show that DIRP consistently breaks existing defense strategies while preserving high image quality, exposing robustness limitations in current mimicry protection techniques. Code is available at `https://github.com/Violette-py/DIRP`.

## 2. Background and Related Work

### 2.1. Text-to-Image Models and Personalization

Personalization techniques further fine-tune T2I models to generate images of a specific subject or style using only a few reference images. For example, DreamBooth (Ruiz et al., 2023) is a representative approach that fine-tunes a diffusion model by associating a rare token (e.g., "[V]") with a target concept, while employing a prior preservation dataset to maintain overall generation quality. In contrast, Textual Inversion (Gal et al., 2022) avoids modifying model weights and instead optimizes a new pseudo-token to represent the target concept. Most personalization methods optimize the diffusion model by minimizing the mean squared error between the predicted noise and the ground-truth noise on the fine-tuning dataset (Ho et al., 2020):

$$\mathcal{L}_{dm}(\theta) = \mathbb{E}_{x_t,t,\epsilon} \left[ \| \epsilon - \epsilon_\theta(x_t, t, c) \|_2^2 \right], \quad (1)$$

where $x_t$ denotes a noisy image, $t$ is a diffusion timestep, and $\epsilon \sim \mathcal{N}(0, I)$ is Gaussian noise. The variable $c$ represents the text condition, typically containing a special token associated with the target concept, and $\epsilon_\theta$ denotes the pre-trained noise prediction network.

### 2.2. Mimicry Protection Methods

**Degradation-based Methods.** Early studies on mimicry protection primarily focused on degradation-based approaches. Inspired by adversarial attacks, these methods

optimize adversarial protective perturbations such that the personalized model exhibits severe generation quality degradation after fine-tuning. Glaze (Shan et al., 2023) leverages style transfer techniques to push the feature representations of protected images toward a predefined target style, thereby inducing diffusion models to establish incorrect style associations during personalization. Mist (Liang et al., 2023) can be viewed as an adversarial attack tailored for diffusion models, where protective perturbations are optimized by maximizing the denoising loss of a pre-trained T2I model. Anti-DreamBooth (Van Le et al., 2023) explicitly simulates the DreamBooth fine-tuning process by constructing a surrogate model and employs projected gradient descent (PGD) (Madry et al., 2017) to generate protective perturbations. MetaCloak (Liu et al., 2024) further introduces a meta-learning framework to address the bilevel optimization problem inherent in adversarial protection, enhancing the transferability and robustness of protective perturbations through an ensemble of surrogate models.

**Detection-based Methods.** Unlike degradation-based approaches, these methods do not completely prevent the model from learning image semantics. Instead, they guide the personalization process to capture a detectable watermark feature, providing a more flexible form of protection. DIAGNOSIS (Wang et al., 2023) introduces subtle geometric distortions into protected images via radial transformations, encouraging the model to memorize these features during personalization and reproduce them in generated images. A subsequent binary classifier is then used to detect the presence of these features for dataset verification and attribution. SIREN (Li et al., 2025) embeds task-relevant coatings into protected images that are specifically aligned with the personalization objective, ensuring that these features can be reliably learned during fine-tuning and remain detectable in the generated outputs.

## 2.3. Perturbation Removal Attacks

Perturbation removal attacks aim to eliminate potential protective perturbations in an image, thereby avoiding quality degradation or the generation of a detectable watermark feature. Representative methods include VAE Attack and Diffusion Attack (Zhao et al., 2024a), which reconstruct images via generative models. This reconstruction process effectively projects images back onto the semantic manifold learned by the model, thereby weakening or even eliminating protective perturbations. Noisy Upscaling (Hönig et al., 2024) can be seen as a variant of Diffusion Attack. It first injects Gaussian noise into the protected image, then employs the Stable Diffusion Upscaler to restore image resolution, mitigating the quality degradation caused by perturbations. IMPRESS (Cao et al., 2023) performs purification by minimizing VAE reconstruction loss. GrIDPure (Zhao et al., 2024b) adopts a patch-wise purification strategy by dividing

the protected image into multiple grids and sequentially purifying each patch using SDEdit. PDM-Pure (Xue & Chen, 2024) demonstrates that pixel-space diffusion models can serve as off-the-shelf purifiers. CAT (Peng et al., 2025) represents a more recent class of perturbation removal attacks, by revealing that the core mechanism of existing protective perturbations is to distort the image distribution in latent space. By adversarially training the VAE encoder to correct the latent-space shift induced by perturbations, CAT effectively neutralizes the protective effects.

## 3. Method

In this section, we compare three reconstruction paradigms through the lens of reconstruction error and show that DDIM inversion uniquely amplifies protective perturbations, leading to severe degradation in reconstructed protected images. Motivated by this observation, we introduce a perturbation removal method that exploits this amplification effect to reveal weaknesses in existing protection mechanisms.

### 3.1. Reconstruction Paradigms

As illustrated in Figure 2, we focus on three representative paradigms in T2I models: *VAE Reconstruction*, *Diffusion Reconstruction*, and *DDIM Inversion Reconstruction*.

Formally, let $x \in \mathbb{R}^{H \times W \times 3}$ denote an input image. We use $\mathcal{E}(\cdot)$ and $\mathcal{D}(\cdot)$ to denote the encoder and decoder of a variational autoencoder (VAE), respectively. In diffusion models, the forward diffusion process progressively corrupts a clean image by injecting Gaussian noise. At timestep $t$, the noisy image can be expressed in closed form as

$$\mathcal{Q}_t(x_0) : x_t = \sqrt{\bar{\alpha}_t}\, x_0 + \sqrt{1 - \bar{\alpha}_t}\, \epsilon, \quad \epsilon \sim \mathcal{N}(0, I), \quad (2)$$

where $\bar{\alpha}_t$ denotes a predefined noise schedule.

The reverse process is parameterized by a trained noise predictor $\epsilon_\theta$. A single DDIM denoising step is defined as

$$\hat{x}_0(x_t) = \frac{x_t - \sqrt{1 - \bar{\alpha}_t}\, \epsilon_\theta(x_t)}{\sqrt{\bar{\alpha}_t}}, \quad (3)$$

$$\mathcal{P}_t(x_t) : x_{t-1} = \sqrt{\bar{\alpha}_{t-1}}\, \hat{x}_0(x_t) + \sqrt{1 - \bar{\alpha}_{t-1}}\, \epsilon_\theta(x_t). \quad (4)$$

Conversely, DDIM inversion reverses the denoising trajectory. The corresponding inversion operator is given by

$$\mathcal{I}_t(x_{t-1}) : x_t = \sqrt{\bar{\alpha}_t}\, \hat{x}_0(x_{t-1}) + \sqrt{1 - \bar{\alpha}_t}\, \epsilon_\theta(x_{t-1}). \quad (5)$$

**VAE Reconstruction.** Given an image, the VAE encoder first maps it to a latent posterior distribution. A deterministic latent representation can be obtained by taking the mean of this posterior, which is then decoded back to the pixel space by the decoder. This process can be formalized as

$$\hat{x}_{vae} = \mathcal{D} \circ \mathcal{E}(x). \quad (6)$$

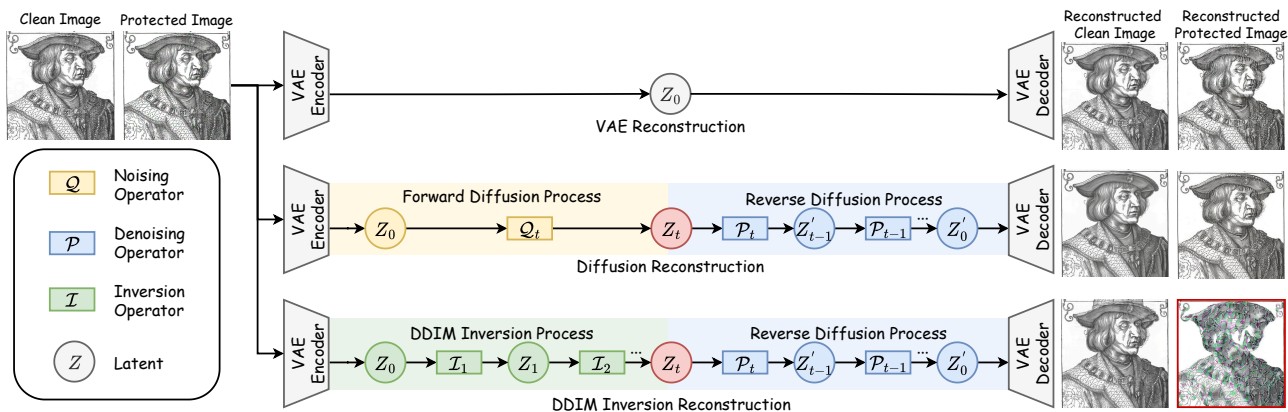

*Figure 2.* Comparison of reconstruction behaviors under different paradigms. DDIM inversion acts as a perturbation amplifier: reconstructions of clean images remain stable, whereas protected images (e.g., Glaze) exhibit severe degradation due to the amplification of off-manifold perturbations.

**Diffusion Reconstruction.** Diffusion reconstruction perturbs the input image into a noisy state through a forward diffusion process and then applies a denoising procedure to recover the image. Since randomness is introduced during the forward noising stage, the resulting reconstructions are non deterministic. This process can be formalized as

$$\hat{x}_{diff} = \mathcal{D} \circ (\mathcal{P}_1 \circ \cdots \circ \mathcal{P}_t) \circ \mathcal{Q}_t \circ \mathcal{E}(x). \tag{7}$$

**DDIM Inversion Reconstruction.** In contrast to the previous two paradigms, DDIM inversion reconstructs an image by explicitly reversing a deterministic DDIM sampling trajectory. As no random noise is introduced during either the inversion or the denoising process, DDIM inversion forms a stable and fully deterministic reconstruction pipeline. This procedure can be expressed as

$$\hat{x}_{inv} = \mathcal{D} \circ (\mathcal{P}_1 \circ \cdots \circ \mathcal{P}_t) \circ (\mathcal{I}_t \circ \cdots \circ \mathcal{I}_1) \circ \mathcal{E}(x). \tag{8}$$

Figure 2 provides a qualitative comparison of reconstruction results obtained by three paradigms on both clean and protected images. For clean inputs, all methods achieve high visual fidelity, indicating their ability to faithfully reconstruct natural images. However, when protection mechanisms are present, the reconstruction behaviors diverge significantly. VAE-based reconstruction and stochastic diffusion reconstruction largely preserve the global appearance of the original content, whereas DDIM inversion reconstruction suffers from pronounced quality degradation and semantic distortion. This property makes DDIM inversion reconstruction act as a perturbation amplifier that is sensitive to protective perturbations, providing a reliable signal for analyzing and revealing the presence of image protection mechanisms.

### 3.2. Reconstruction Error Analysis

To understand why different reconstruction mechanisms respond differently to protective perturbations, we formally analyze image reconstruction error through the lens of diffusion dynamics. In *VAE reconstruction*, the process primarily functions as a projection within the latent space. The bottleneck architecture of the encoder-decoder pair acts as an implicit regularizer, capable of absorbing low-magnitude, structured deviation. Similarly, *diffusion reconstruction* injects Gaussian noise during the forward process. While this stochasticity facilitates generative diversity, it acts as a low-pass filter that disrupts the precise phase and structural correspondence of latent perturbations. This effectively "washes away" subtle protective signals, explaining the prevalence of VAE and stochastic diffusion as baseline tools for perturbation purification (Zhao et al., 2024a).

In contrast, *DDIM Inversion* enforces a deterministic trajectory, constraining the reconstruction to adhere to the manifold implicitly defined by the model's score function. While this ensures high-fidelity cycle consistency for natural images conforming to the training distribution (Chung et al., 2022; Mokady et al., 2023), it creates a lack of robustness for off-manifold samples. When an input deviates from the learned distribution, these deviations cannot be locally absorbed; instead, they accumulate along the deterministic inversion trajectory, manifesting as severe semantic distortions. Consequently, DDIM Inversion acts as a perturbation amplifier, where deviations are amplified into salient reconstruction artifacts.

To formalize this amplification effect, we define the single-step latent reconstruction error at timestep $t$. By considering the composition of the inversion operator $\mathcal{I}$ and the denoising operator $\mathcal{P}$, the residual $\Delta z_t$ can be expressed as:

$$\Delta z_t = \gamma_t \left[ \epsilon_\theta(z_t) - \epsilon_\theta(\mathcal{I}_{t+1}(z_t)) \right] \tag{9}$$

where $\gamma_t = \sqrt{\frac{\bar{\alpha}_t(1-\bar{\alpha}_{t+1})}{\bar{\alpha}_{t+1}}} - \sqrt{1-\bar{\alpha}_t}$ is a coefficient determined by the noise schedule. A comprehensive derivation

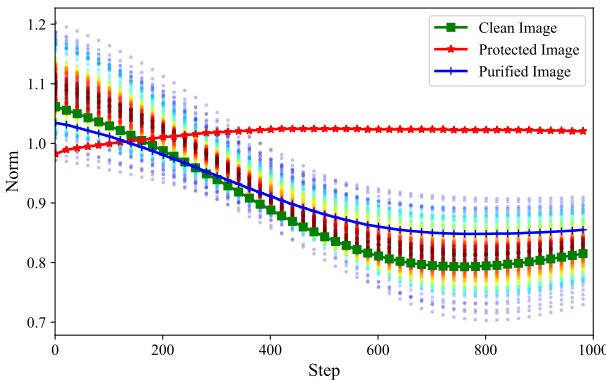

*Figure 3.* Manifold density distributions during DDIM inversion. Protected images (e.g., Anti-DreamBooth) deviate significantly from the normal high-density regions, while DIRP effectively purifies these samples and pulls these samples back toward the normal manifold.

of this residual is provided in Appendix A.

Assuming the noise predictor $\epsilon_\theta$ is locally differentiable and the inversion displacement $\delta z_t = \mathcal{I}_{t+1}(z_t) - z_t$ is sufficiently small, we apply a first-order Taylor expansion to linearize the error:

$$\Delta z_t \approx -\gamma_t \, \mathbf{J}_{\epsilon_\theta}(z_t) \, \delta z_t, \quad \mathbf{J}_{\epsilon_\theta}(z_t) = \nabla_z \epsilon_\theta(z_t) \qquad (10)$$

This indicates that the reconstruction error magnitude is fundamentally governed by the spectral properties of the noise predictor's Jacobian, $\mathbf{J}_{\epsilon_\theta}$. Notably, existing protection methods (e.g., Mist, Anti-DreamBooth) typically optimize perturbations to maximize fine-tuning gradients. Mathematically, this is equivalent to artificially inflating the spectral norm of $\mathbf{J}_{\epsilon_\theta}$ in specific directions within the latent space, thereby ensuring that even minute displacements $\delta z_t$ result in significant reconstruction residuals. Further analysis on this spectral inflation is detailed in Appendix B.

Finally, we demonstrate that this mechanism yields a larger error for protected images than for clean images. Let $\Delta z_t^{\mathrm{cl}}$ and $\Delta z_t^{\mathrm{pr}}$ denote the single-step latent reconstruction errors for clean and protected samples, respectively. As illustrated in Figure 3, for clean images, the latent state $z_t^{\mathrm{cl}}$ resides in high-density regions of the manifold where $\epsilon_\theta$ is locally smooth and $\delta z_t$ is minimized. Conversely, protected images introduce structured perturbations that cause semantic drift, simultaneously enlarging the inversion displacement and situating the latent in high-curvature regions of the score field. Under these conditions, the inequality $\|\Delta z_t^{\mathrm{pr}}\|_2 > \|\Delta z_t^{\mathrm{cl}}\|_2$ holds across the diffusion trajectory, providing a theoretical explanation for the progressive amplification of protective perturbations during DDIM inversion. Detailed proofs and distributional assumptions are provided in Appendix C.

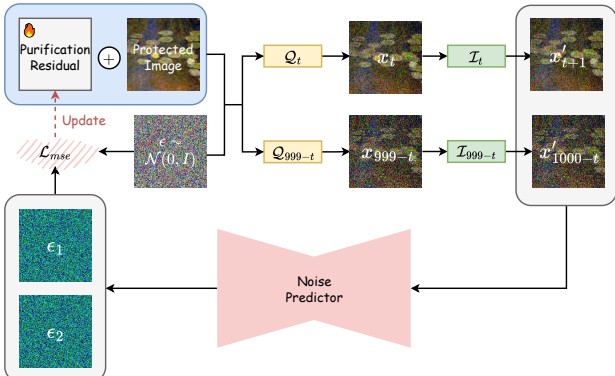

*Figure 4.* Overview of DIRP.

### 3.3. DDIM Inversion-based Reconstruction Purification

Based on the preceding analysis, we identify that an effective purification mechanism must satisfy two interdependent criteria. The first is *reconstruction consistency*: the purified image must align with the manifold learned by the diffusion model, such that it satisfies the deterministic cycle-consistency of DDIM Inversion. The second is *perceptual fidelity*: the purification operation must preserve the original visual attributes in both pixel and perceptual spaces, ensuring the purified result is indistinguishable from the original to the human visual system.

Guided by these principles, we propose *DDIM Inversion-based Reconstruction Purification (DIRP)*. As illustrated in Figure 4, given a protected image $\tilde{x}$, DIRP learns a purification residual $\delta$, yielding a purified image $x_p = \tilde{x} + \delta$. The primary objective is to project $x_p$ back onto the learned semantic manifold, thereby restoring the model's capacity to represent the underlying structural features during subsequent personalization fine-tuning.

**Optimization Objective.** To jointly enforce reconstruction consistency and perceptual similarity, we formulate perturbation purification as a constrained optimization problem. For reconstruction consistency, we directly leverage the DDIM Inversion single-step latent reconstruction error $\Delta z_t$ derived in Equation 9 earlier as the optimization signal, minimizing it to suppress instability in the inversion process. For perceptual similarity, we impose an $\ell_\infty$ norm constraint on the purification perturbation in pixel space, limiting its magnitude to an imperceptible range. Overall, the objective can be summarized as minimizing the DDIM reconstruction error under a perceptual constraint:

$$\mathcal{L} = \min_{\|\delta\|_\infty \le \varepsilon} \left\| \epsilon_\theta(z_t^\delta) - \epsilon_\theta\big(\mathcal{I}_{t+1}(z_t^\delta)\big) \right\|_2^2, \qquad (11)$$

where $z_t^\delta$ denotes the latent representation of the image at timestep $t$, and $\mathcal{I}_{t+1}$ is the DDIM inversion operator. The $\ell_\infty$-norm constraint ensures perceptual similarity.

**Semantic Shortcut (SS).** For any sample $x_p$ conforming to the training distribution, the noise predictor should ideally satisfy $\epsilon_\theta(z_t) \approx \epsilon$, where $\epsilon$ is the ground-truth Gaussian noise injected at timestep $t$. By substituting the model's potentially biased prediction $\epsilon_\theta(z_t^\delta)$ with the true noise $\epsilon$, we provide a more stable and distribution-aware supervision signal. This steers the optimization toward noise structures that are semantically consistent with the diffusion prior. The resulting objective is formulated as:

$$\min_{\|\delta\|_\infty \leq \varepsilon} \mathcal{L} = \left\| \epsilon - \epsilon_\theta\big(\mathcal{I}_{t+1}(z_t^\delta)\big) \right\|_2^2. \qquad (12)$$

**Bidirectional Asynchronous Optimization (BAC).** In practice, the magnitude and variance of the noise prediction error vary significantly across the diffusion trajectory. This temporal imbalance often leads to gradient oscillations. To stabilize the optimization, we propose a *Bidirectional Asynchronous Optimization* strategy. In each iteration, we jointly optimize the reconstruction loss at two symmetric timesteps, $t$ and $T - t - 1$:

$$\mathcal{L}_{\text{total}} = \min_{\|\delta\|_\infty \leq \varepsilon} \frac{1}{2} \sum_{k \in \{t, T-t-1\}} \left\| \epsilon - \epsilon_\theta\big(\mathcal{I}_{k+1}(z_k^\delta)\big) \right\|_2^2,$$
$$(13)$$

by coupling early-stage and late-stage diffusion dynamics, this symmetric scheduling balances the gradient contributions across different temporal scales, significantly reducing optimization variance and accelerating convergence.

**Latent Residual Refinement (LRR).** While the DIRP objective effectively mitigates errors introduced by the noise predictor and the inversion operator $\mathcal{I}$, it does not explicitly account for mapping residuals inherent in the VAE encoder-decoder architecture. In cases where high-frequency protection artifacts persist, we optionally apply a lightweight refinement step, such as AE denoising. As demonstrated in our ablation studies (subsection 4.2), while LRR is insufficient to bypass mimicry protection on its own, its integration with DIRP further enhances the overall quality and stability of the purified results, achieving a better balance between perceptual fidelity and semantic consistency.

## 4. Experiments

### 4.1. Experimental Settings

**Personalization Methods and Baselines.** We evaluate the performance of DIRP under three mainstream personalization settings, namely LoRA fine-tuning (Hu et al., 2022), DreamBooth (Ruiz et al., 2023), and Textual Inversion (Gal et al., 2022). Additional fine-tuning details are provided in Appendix E. We consider six SOTA mimicry protection methods, including four degradation-based approaches, Glaze (Shan et al., 2023), Mist (Liang et al., 2023), Anti-DreamBooth (Van Le et al., 2023), and MetaCloak (Liu

et al., 2024), as well as two detection-based approaches, DI-AGNOSIS (Wang et al., 2023) and SIREN (Li et al., 2025). To comprehensively assess the effectiveness of DIRP in removing protective perturbations, we compare it against seven representative perturbation removal attacks: Diffusion Attack (DA) (Nie et al., 2022; Zhao et al., 2024a), VAE Attack (VA) (Zhao et al., 2024a), IMPRESS (Cao et al., 2023), Noisy Upscaling (NU) (Hönig et al., 2024), CAT (Peng et al., 2025), GrIDPure (Zhao et al., 2024b) and PDM-Pure (Xue & Chen, 2024). All baseline methods strictly follow the default configurations in their official implementations. Further implementation details are provided in Appendix F.

**Models and Datasets.** We adopt Stable Diffusion v2.1-base (SD-v2.1) (AI, 2022) as the default surrogate model for mimicry protection methods as well as the fine-tuning model in the personalization stage. It is worth noting that, under this setting, protective perturbations are optimized against the target model in a white-box manner and therefore represent an upper bound on the effectiveness of mimicry protection methods. To evaluate the cross-model generalization of DIRP, we further include additional models, such as Flux.1-dev (Labs), in our ablation studies. For datasets, we conduct experiments on two widely used benchmarks, WikiArt (Saleh & Elgammal, 2015) and the Pokémon dataset (Kaggle, 2019). For each concept, we randomly select four related images as fine-tuning data for subsequent personalization. Since Glaze is specifically designed for style imitation and SIREN only provides an open-source watermark encoder for the Pokémon dataset, we evaluate the degradation-based methods on WikiArt and the detection-based methods on the Pokémon dataset. Additional dataset details are provided in Appendix D.

**Evaluation Metrics.** To evaluate the amplification effect of DDIM inversion reconstruction, we measure the quality degradation of protected images before and after reconstruction using SSIM (Wang et al., 2004), PSNR (Hore & Ziou, 2010), LPIPS (Zhang et al., 2018), and DINO (Oquab et al., 2023). For assessing protective perturbations removal performance, we evaluate by generating 500 new images for each concept using the personalized models, and use 500 corresponding real images as reference sets to support metric computation. For degradation-based mimicry protection methods, we measure overall image quality using the Fréchet Inception Distance (FID) (Heusel et al., 2017) and evaluate semantic consistency between generated and reference images via cosine similarity (CS) computed from a CLIP-based image encoder (Radford et al., 2021). Lower FID indicates higher generation quality, while higher CS reflects better semantic preservation of the original concept, signaling effective removal of protective perturbations. For detection-based mimicry protection methods, in addition to these perceptual quality metrics, we report the Watermark Accuracy (WACC) as a robustness measure.

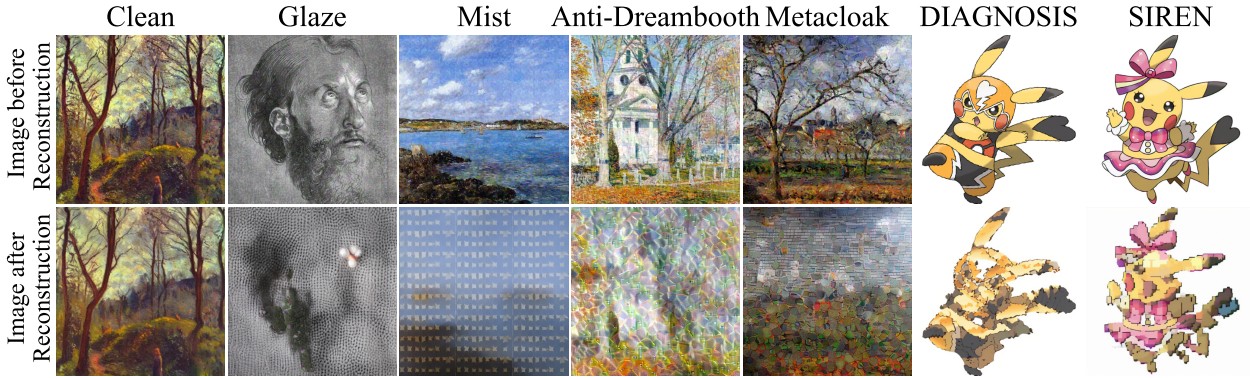

*Figure 5.* [RQ1] Qualitative comparison of clean and protected images in the DDIM inversion reconstruction results.

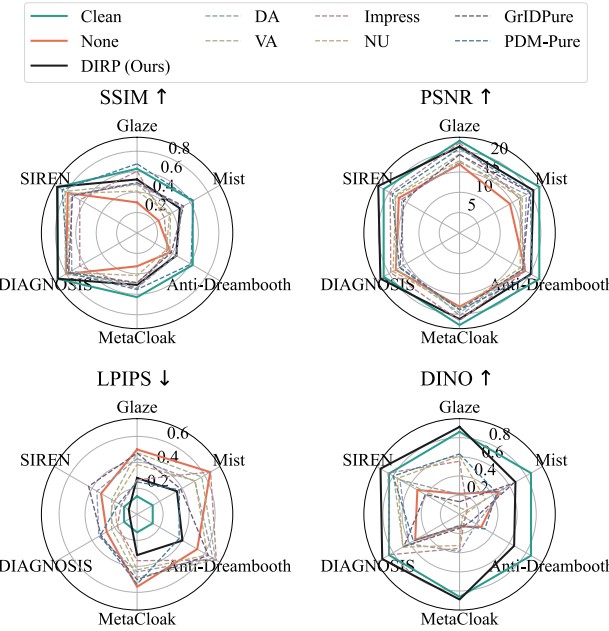

*Figure 6.* [RQ1] Quantitative reconstruction fidelity (SSIM, PSNR, LPIPS, DINO) across various protection methods.

## 4.2. Experimental Results

Our extensive experiments aim to answer the following research questions (RQs):

- [RQ1] How does DIRP perform in improving the quality of DDIM inversion reconstruction?

- [RQ2] How effective is DIRP in removing protective perturbations?

- [RQ3] How do different modules affect the performance of DIRP?

**RQ1: Effect of DIRP on DDIM Inversion Reconstruction Quality.** Since the textual descriptions associated with images are typically unknown or unavailable in real-world

scenarios, we adopt an empty prompt in all experiments and perform DDIM inversion with 50 timesteps to reconstruct both clean and protected images. As shown in Figure 5, for unprotected clean images, DDIM inversion stably maps the inputs back onto the semantic manifold learned by the model, yielding reconstructions that are highly consistent with the originals in both global semantics and structural content. In contrast, for images protected by either degradation-based or detection-based methods, the embedded protective perturbations are substantially amplified during the DDIM inversion process, resulting in pronounced structural distortions in the reconstructed images. This behavior closely aligns with our analysis: the protective perturbations gradually accumulate and are amplified, severely undermining the stability of DDIM inversion reconstruction.

Figure 6 reports the corresponding quantitative results, which further corroborate these observations and demonstrate that DIRP achieves SOTA performance in improving DDIM inversion reconstruction quality. Beyond comparing clean images (Clean) and unpurified protected images (None), we evaluate six perturbation removal attacks. Three key findings can be drawn from the results. First, protected images exhibit significantly worse performance than clean images across all four metrics providing quantitative evidence for the perturbation amplification effect of DDIM inversion on protective perturbations. Second, while existing perturbation removal attacks partially alleviate reconstruction degradation and improve upon the None baseline, a substantial gap with respect to the Clean setting remains, indicating their limited ability to effectively eliminate protective perturbations. Finally, DIRP consistently and effectively purifies protective perturbations introduced by all protection methods, enabling the reconstructed images to approach the quality of clean reconstructions across all evaluation metrics. These results highlight that, by directly targeting DDIM inversion reconstruction error as a fundamental signal, DIRP can precisely remove protective perturbations and thereby preserve the fidelity of the reconstruction process.

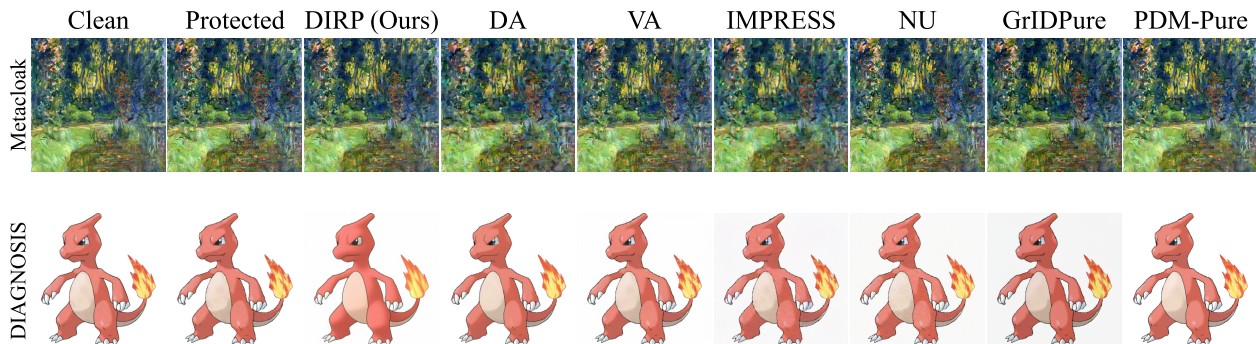

*Figure 7.* [RQ2] Qualitative comparison of the purified image and the protected image.

*Table 1.* [RQ2] Performance on Textual Inversion. **Bold** values indicate best performance; *underlined italic* values indicate second best.

| | Glaze | | Mist | | Anti-DreamBooth | | Metacloak | | DIAGNOSIS | | | SIREN | | |
|---|---|---|---|---|---|---|---|---|---|---|---|---|---|---|
| | FID | CS | FID | CS | FID | CS | FID | CS | WACC | FID | CS | WACC | FID | CS |
| None | 196.66 | 0.738 | 357.84 | 0.687 | 382.77 | 0.629 | 262.95 | 0.704 | 99.90% | 153.55 | 0.774 | 49.70% | **121.89** | 0.795 |
| DA | 231.26 | 0.705 | 367.59 | 0.686 | 347.84 | 0.645 | 237.68 | 0.717 | *94.10%* | **132.39** | 0.775 | **0.00%** | 129.27 | **0.813** |
| VA | 185.09 | *0.814* | *211.45* | 0.759 | 259.38 | 0.702 | *217.48* | *0.758* | 99.60% | 169.35 | 0.777 | 3.87% | 147.29 | *0.805* |
| IMPRESS | *183.16* | 0.741 | 356.26 | 0.691 | 282.83 | 0.671 | 247.39 | 0.721 | 98.90% | 157.47 | **0.785** | 24.40% | 142.23 | 0.768 |
| NU | 190.30 | **0.822** | 240.83 | *0.777* | *236.42* | 0.744 | 222.20 | 0.753 | 99.80% | 149.06 | **0.785** | 32.85% | 123.58 | 0.770 |
| CAT | 209.20 | 0.718 | 339.22 | 0.697 | 355.47 | 0.626 | 296.43 | 0.693 | 99.80% | 146.07 | 0.761 | 35.45% | 126.16 | 0.794 |
| GrIDPure | 189.56 | 0.800 | 218.78 | 0.770 | 256.95 | 0.693 | 239.48 | 0.719 | 99.80% | 146.49 | **0.785** | 0.24% | *123.35* | 0.787 |
| PDM-Pure | 223.16 | 0.780 | 223.83 | 0.772 | 237.56 | *0.760* | 245.37 | 0.746 | 96.20% | 145.54 | 0.767 | *0.01%* | 148.14 | 0.786 |
| DIRP (Ours) | **174.64** | 0.812 | **191.80** | **0.785** | **197.79** | **0.787** | **183.31** | **0.783** | **1.80%** | *135.45* | *0.783* | 0.10% | 134.39 | 0.776 |

**RQ2: Effectiveness of DIRP in Removing Protective Perturbations.** We next evaluate the utility of DIRP for downstream personalization tasks. We first compare the purified images produced by different purification attacks, which are directly used as training data for personalized fine-tuning. As shown in Figure 7, we visualize the purification results on Metacloak and DIAGNOSIS, together with clean and protected images for qualitative comparison. Similar to baseline attacks, DIRP alleviates the protective perturbations introduced by degradation-based methods and restores more natural visual structures. However, for DIAGNOSIS, we observe a clear distinction: only DIRP effectively removes the edge distortions and recovers smooth image boundaries comparable to those of clean images. In contrast, baseline purification methods still retain noticeable jagged watermark-like artifacts after purification, which likely explains their failure to suppress DIAGNOSIS detection.

Table 1 reports the results for Textual Inversion fine-tuning. Across all six mimicry protection methods and evaluation metrics, DIRP consistently achieves the best or second-best performance in both generation quality and semantic fidelity. For degradation-based protection methods, DIRP significantly reduces FID while improving CS, indicating that purified images enable the model to relearn the underlying concepts. For detection-based protection methods, the limitations of prior attacks are even more pronounced.

For example, with DIAGNOSIS, all baseline attacks fail to suppress the embedded watermark, with WACC remaining above 90% in nearly all cases. In comparison, DIRP reduces DIAGNOSIS WACC to below 2% and SIREN WACC nearly to zero, while maintaining competitive FID and CS. These results demonstrate the broad applicability of DIRP: it effectively neutralizes both degradation-based and detection-based protections, whereas existing attacks are primarily limited to degradation-based mechanisms. Results on DreamBooth and LoRA exhibit the same trend and are reported in Appendix G for completeness.

Figure 8 further visualizes these effects. The first row shows images generated from unpurified protected data. For degradation-based protections, generated images exhibit pronounced artifacts or structural collapse, confirming the effectiveness of these defenses. For detection-based protections, overall image quality remains relatively high, but DIAGNOSIS introduces subtle distortions, reflecting the successful embedding of detectable watermark features. The second row shows images generated after applying DIRP. Across all protection methods, image quality improves substantially. Notably, for DIAGNOSIS, characteristic distortions nearly disappear, yielding smooth, natural outputs. This visual improvement aligns with the sharp reduction in watermark detection accuracy, confirming that DIRP effectively removes protective perturbations.

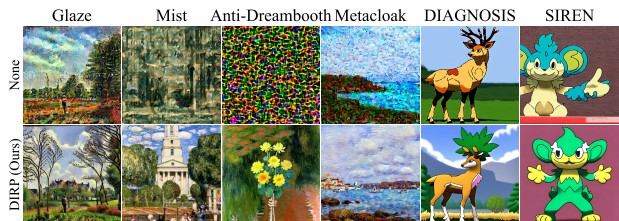

*Figure 8.* [RQ2] Personalized generation results using Textual Inversion. DIRP eliminates the structural collapse and characteristic artifacts induced by degradation- and detection-based protections.

**RQ3: Contribution of Individual Modules in DIRP.** We conduct an ablation study to quantify the contribution of each component in DIRP by removing the Semantic Shortcut (SS), Bidirectional Asynchronous Optimization (BAC), and Latent Residual Refinement (LRR). The DreamBooth fine-tuning results on Metacloak and DIAGNOSIS are summarized in Table 2.

Removing SS causes a severe degradation across all metrics, indicating that semantically grounded and distribution-consistent supervision is essential for projecting purified images back onto the learned diffusion manifold. Excluding BAC leads to a noticeable but milder performance drop, with a pronounced increase in DIAGNOSIS WACC, suggesting that imbalanced gradient contributions across diffusion timesteps hinder effective suppression of protective perturbations. Omitting LRR results in increased residual artifacts and degraded generation quality. Notably, LRR alone reduces DIRP to a VAE attack, which fails to suppress DIAGNOSIS detection, demonstrating that LRR is ineffective without DIRP. Overall, the complete DIRP framework consistently achieves the best performance. SS provides semantically consistent supervision, BAC stabilizes optimization across diffusion timesteps, and LRR mitigates residual artifacts. Their synergy is crucial for robust perturbation removal while preserving perceptual quality and semantic fidelity, validating the design choices of DIRP.

We further evaluate cross-architecture generalization using Flux.1-dev, whose architecture differs from the SD-v2.1 model used to optimize Metacloak perturbations. Unpurified images achieve relatively strong FID and CS, reflecting the limited transferability of degradation-based perturbations across architectures. In contrast, DIAGNOSIS maintains a WACC of 100%, indicating that it remains highly effective on unseen models. After applying DIRP, DIAGNOSIS WACC drops to 0%, while perceptual and semantic metrics improve, confirming that DIRP robustly removes protective perturbations even under cross-model settings.

Furthermore, we conduct a detailed analysis of the impact of hyperparameters on DIRP, as shown in Table 3. Overall, DIRP exhibits robustness to hyperparameter choices, main-

*Table 2.* [RQ3] Ablation study of our DIRP.

|  | Metacloak | | DIAGNOSIS | | |
| --- | --- | --- | --- | --- | --- |
|  | FID ↓ | CS ↑ | WACC ↓ | FID ↓ | CS ↑ |
| DIRP (Ours) | **212.71** | 0.828 | **30.80%** | 127.46 | 0.880 |
| w/o SS | 320.05 | 0.686 | 100.00% | 198.61 | 0.719 |
| w/o BAC | 221.34 | **0.837** | 45.40% | **126.12** | **0.884** |
| w/o LRR | 217.18 | 0.720 | 81.00% | 130.54 | 0.883 |
| Flux.1-dev + None | 218.51 | 0.837 | 99.90% | 133.25 | **0.906** |
| Flux.1-dev + DIRP | **195.08** | **0.847** | **0.80%** | **119.17** | 0.897 |

*Table 3.* [RQ3] Impact of the number of epochs and perturbation budget. The default settings are epoch $= 500$ and $\epsilon = 24/255$.

| | Metacloak | | | Metacloak | |
| --- | --- | --- | --- | --- | --- |
| Epoch | FID ↓ | CS ↑ | Budget | FID ↓ | CS ↑ |
| 100 | 217.67 | 0.825 | $\epsilon$=8/255 | 236.40 | 0.808 |
| 300 | 200.93 | 0.837 | $\epsilon$=16/255 | 217.98 | 0.827 |
| 500 | 212.71 | 0.828 | $\epsilon$=24/255 | 212.71 | 0.828 |
| 700 | 214.70 | 0.809 | $\epsilon$=32/255 | 213.71 | 0.802 |

taining stable and competitive performance across a wide range of epochs from 100 to 700. As the number of training epochs increases, the performance gain gradually saturates, while excessively large epochs lead to slight degradation, which may be attributed to overfitting. This issue can be mitigated by appropriately selecting the number of training epochs or introducing regularization and early-stopping strategies.

On the other hand, the perturbation budget $\epsilon$ introduces a natural trade-off between purification strength and image fidelity: a smaller budget may be insufficient to fully remove protective perturbations, whereas an overly large budget can compromise semantic consistency. Notably, existing protection methods typically rely on small budgets (e.g., $16/255$) to maintain imperceptibility, while our results demonstrate that DIRP remains effective in purification and preserves high semantic fidelity even under same budget regimes.

## 5. Conclusion

In this work, we identify a previously overlooked perturbation amplification effect in DDIM inversion, showing that imperceptible protective perturbations can lead to large reconstruction errors when images deviate from the learned semantic manifold. Based on this observation, we propose DIRP, which removes protective perturbations by minimizing DDIM inversion reconstruction error under perceptual constraints. Extensive experiments demonstrate that DIRP consistently outperforms existing removal attacks and exposes critical robustness limitations in current mimicry protection methods, highlighting the need for more principled and robust defenses.

## Acknowledgements

We would like to thank the anonymous reviewers for their insightful comments that helped improve the quality of the paper. This work was supported in part by the National Natural Science Foundation of China (62472096, 62502157, U23B2021). Min Yang is a faculty of Shanghai Pudong Research Institute of Cryptology, and Engineering Research Center of Cyber Security Auditing and Monitoring, Ministry of Education, China. Mi Zhang and Min Yang are the corresponding authors.

## Impact Statement

This work introduces DIRP and provides the first systematic demonstration of the vulnerability of existing mimicry protection methods under DDIM inversion reconstruction. While the techniques studied could, if misused, pose potential risks, our primary goal is to deepen the understanding of current protection methods and offer a new evaluation perspective for designing more robust and reliable personalization workflows. Overall, we believe this work contributes to the development of mimicry protection technologies and, ultimately, supports the creation of more robust safeguards for protected images.

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

# A. Derivation of Single-Step DDIM Inversion Reconstruction Error

In this section, we provide a detailed derivation of the single-step latent reconstruction error under the deterministic DDIM framework. Our goal is to show that the discrepancy between an initial latent $z_t$ and its reconstructed counterpart $z'_t$ after one inversion-denoising cycle is governed by the variation in the noise predictor $\epsilon_\theta$.

## A.1. Preliminaries

Consider a deterministic DDIM trajectory. Let $z_t$ denote the latent at timestep $t$. The inversion operator $\mathcal{I}_{t+1}$ maps $z_t$ to the next timestep $t + 1$, and the denoising operator $\mathcal{P}_t$ maps a latent from $t + 1$ back to $t$.

The predicted $\hat{z}_0$ at any timestep $t$ is given by

$$\hat{z}_0(z_t) = \frac{z_t - \sqrt{1 - \bar{\alpha}_t}\, \epsilon_\theta(z_t)}{\sqrt{\bar{\alpha}_t}}. \tag{14}$$

## A.2. Forward Inversion Step

The DDIM inversion from timestep $t$ to $t + 1$ is defined as

$$z_{t+1} = \mathcal{I}_{t+1}(z_t) = \sqrt{\bar{\alpha}_{t+1}}\, \hat{z}_0(z_t) + \sqrt{1 - \bar{\alpha}_{t+1}}\, \epsilon_\theta(z_t). \tag{15}$$

Substituting (14) into (15), we can rewrite $z_{t+1}$ in terms of $z_t$:

$$z_{t+1} = \sqrt{\frac{\bar{\alpha}_{t+1}}{\bar{\alpha}_t}}\, z_t + \left( \sqrt{1 - \bar{\alpha}_{t+1}} - \sqrt{\frac{\bar{\alpha}_{t+1}(1 - \bar{\alpha}_t)}{\bar{\alpha}_t}} \right) \epsilon_\theta(z_t). \tag{16}$$

## A.3. Reverse Denoising Step

The subsequent reconstruction step $\mathcal{P}_t$ maps $z_{t+1}$ back to timestep $t$:

$$z'_t = \mathcal{P}_t(z_{t+1}) = \sqrt{\bar{\alpha}_t}\, \hat{z}_0(z_{t+1}) + \sqrt{1 - \bar{\alpha}_t}\, \epsilon_\theta(z_{t+1}). \tag{17}$$

Substituting (14) for $\hat{z}_0(z_{t+1})$, we have

$$z'_t = \sqrt{\frac{\bar{\alpha}_t}{\bar{\alpha}_{t+1}}}\, z_{t+1} + \left( \sqrt{1 - \bar{\alpha}_t} - \sqrt{\frac{\bar{\alpha}_t(1 - \bar{\alpha}_{t+1})}{\bar{\alpha}_{t+1}}} \right) \epsilon_\theta(z_{t+1}). \tag{18}$$

## A.4. Deriving the Single-Step Error

The reconstruction error is defined as

$$\Delta z_t = z'_t - z_t.$$

Substituting (16) into (18) and simplifying the coefficients yields

$$z'_t = z_t + \underbrace{\left( \sqrt{\frac{\bar{\alpha}_t(1 - \bar{\alpha}_{t+1})}{\bar{\alpha}_{t+1}}} - \sqrt{1 - \bar{\alpha}_t} \right)}_{\gamma_t} [\epsilon_\theta(z_t) - \epsilon_\theta(z_{t+1})]. \tag{19}$$

Thus, the single-step latent reconstruction error can be compactly expressed as

$$\Delta z_t = \gamma_t \left[ \epsilon_\theta(z_t) - \epsilon_\theta(z_{t+1}) \right], \quad z_{t+1} = \mathcal{I}_{t+1}(z_t), \tag{20}$$

where the scalar $\gamma_t$ depends on the diffusion schedule:

$$\gamma_t = \sqrt{\frac{\bar{\alpha}_t(1 - \bar{\alpha}_{t+1})}{\bar{\alpha}_{t+1}}} - \sqrt{1 - \bar{\alpha}_t}. \tag{21}$$

## B. First-Order Linearization of Reconstruction Error

In this section, we provide the formal derivation for the first-order approximation of the single-step latent reconstruction error. This analysis reveals how the local geometry of the noise predictor $\epsilon_\theta$, characterized by its Jacobian matrix, dictates the amplification of off-manifold perturbations.

### B.1. Exact Error Formulation

As derived in the preceding analysis, the single-step latent reconstruction error $\Delta z_t = z'_t - z_t$ for a deterministic DDIM inversion-denoising cycle is given by:

$$\Delta z_t = \gamma_t \Big[ \epsilon_\theta(z_t) - \epsilon_\theta(z_{t+1}) \Big] \tag{22}$$

where $z_{t+1} = \mathcal{I}_{t+1}(z_t)$ is the latent obtained via the DDIM inversion operator, and the scalar coefficient $\gamma_t$ is defined based on the noise schedule:

$$\gamma_t = \sqrt{\frac{\bar{\alpha}_t(1 - \bar{\alpha}_{t+1})}{\bar{\alpha}_{t+1}}} - \sqrt{1 - \bar{\alpha}_t}. \tag{23}$$

### B.2. First-Order Taylor Expansion

To analyze the sensitivity of this error to latent perturbations, we assume that the noise predictor $\epsilon_\theta$ is locally differentiable in the neighborhood of $z_t$. Let $\delta z_t$ denote the latent displacement induced by the inversion operator:

$$\delta z_t = z_{t+1} - z_t. \tag{24}$$

Applying a first-order Taylor expansion to the term $\epsilon_\theta(z_{t+1})$ centered at $z_t$, we obtain:

$$\epsilon_\theta(z_{t+1}) = \epsilon_\theta(z_t + \delta z_t) = \epsilon_\theta(z_t) + \mathbf{J}_{\epsilon_\theta}(z_t)\delta z_t + \mathcal{O}(\|\delta z_t\|^2) \tag{25}$$

where $\mathbf{J}_{\epsilon_\theta}(z_t) = \nabla_z \epsilon_\theta(z_t) \in \mathbb{R}^{d \times d}$ is the Jacobian matrix of the noise predictor with respect to the latent state at timestep $t$.

### B.3. Jacobian-Mediated Error Approximation

Substituting the expansion from (25) into the exact error formula (22):

$$\begin{aligned} \Delta z_t &= \gamma_t \Big[ \epsilon_\theta(z_t) - \big( \epsilon_\theta(z_t) + \mathbf{J}_{\epsilon_\theta}(z_t)\delta z_t + \mathcal{O}(\|\delta z_t\|^2) \big) \Big] \\ &= -\gamma_t \mathbf{J}_{\epsilon_\theta}(z_t)\delta z_t - \gamma_t \mathcal{O}(\|\delta z_t\|^2). \end{aligned} \tag{26}$$

By neglecting high-order terms for sufficiently small inversion steps, we arrive at the linear approximation:

$$\Delta z_t \approx -\gamma_t \mathbf{J}_{\epsilon_\theta}(z_t)\delta z_t. \tag{27}$$

## C. Comparative Analysis of Reconstruction Error Magnitudes

In this section, we provide a comparative analysis to prove that the latent reconstruction error for protected images is strictly larger than that for clean images, i.e., $\|\Delta z_t^{\mathrm{pr}}\|_2 > \|\Delta z_t^{\mathrm{cl}}\|_2$. This derivation utilizes the first-order approximation established in Appendix B and incorporates the geometric properties of the learned semantic manifold.

### C.1. Error Magnitude Formulation

From (27), the magnitude of the single-step reconstruction error at timestep $t$ is given by:

$$\|\Delta z_t\|_2 \approx |\gamma_t| \cdot \|\mathbf{J}_{\epsilon_\theta}(z_t)\delta z_t\|_2, \tag{28}$$

where $\delta z_t = z_{t+1} - z_t$ represents the inversion displacement. Applying the properties of matrix norms, we have the following lower bound for the error:

$$\|\Delta z_t\|_2 \gtrsim |\gamma_t| \cdot \sigma_{\min}(\mathbf{J}_{\epsilon_\theta}(z_t)) \cdot \|\delta z_t\|_2, \tag{29}$$

and an upper bound characterized by the spectral norm (the largest singular value $\sigma_{\max}$):

$$\|\Delta z_t\|_2 \lesssim |\gamma_t| \cdot \sigma_{\max}(\mathbf{J}_{\epsilon_\theta}(z_t)) \cdot \|\delta z_t\|_2. \tag{30}$$

### C.2. Analysis for Clean Images

For a clean image $x_{\mathrm{cl}}$, the latent $z_t^{\mathrm{cl}}$ lies on or in close proximity to the high-density semantic manifold $\mathcal{M}$ learned during the diffusion model's training. We characterize this state by two key properties:

1. **Local Score Smoothness:** On the manifold $\mathcal{M}$, the noise predictor $\epsilon_\theta$ (which approximates the score function $\nabla_z \log p_t(z)$) is well-conditioned and locally smooth. This implies that the spectral norm of its Jacobian is bounded by a relatively small constant $K_{\mathrm{cl}}$:

$$\sigma_{\max}(\mathbf{J}_{\epsilon_\theta}(z_t^{\mathrm{cl}})) \leq K_{\mathrm{cl}}. \tag{31}$$

2. **Trajectory Consistency:** Since the deterministic DDIM ODE is optimized for data following the training distribution, the inversion displacement $\|\delta z_t^{\mathrm{cl}}\|_2$ remains minimal, as the trajectory is nearly perfectly reversible for on-manifold samples.

### C.3. Analysis for Protected Images

Protected images $\tilde{x}_{\mathrm{pr}}$ are generated by injecting adversarial perturbations $\delta$ such that the resulting latent $\tilde{z}_t^{\mathrm{pr}}$ is pushed into off-manifold regions. Existing mimicry protection methods (e.g., Mist, Anti-DreamBooth) explicitly optimize $\delta$ to maximize the model's loss, which yields:

1. **Spectral Norm Inflation:** The protection optimization process is equivalent to maximizing the local curvature of the energy landscape in latent space. This leads to a significant increase in the sensitivity of the noise predictor, resulting in a large spectral norm:

$$\sigma_{\max}(\mathbf{J}_{\epsilon_\theta}(\tilde{z}_t^{\mathrm{pr}})) = K_{\mathrm{pr}}, \quad \text{where } K_{\mathrm{pr}} \gg K_{\mathrm{cl}}. \tag{32}$$

2. **Inversion Misalignment:** Because the latent $\tilde{z}_t^{\mathrm{pr}}$ exhibits semantic inconsistency with the learned manifold, the deterministic inversion operator $\mathcal{I}_{t+1}$ struggles to map the sample to a stable Gaussian representation. This causes a larger inversion displacement $\|\delta \tilde{z}_t^{\mathrm{pr}}\|_2$ compared to clean samples.

### C.4. Formal Proof of Error Disparity

Given the relationships established above, we compare the error magnitudes:

$$\frac{\|\Delta z_t^{\mathrm{pr}}\|_2}{\|\Delta z_t^{\mathrm{cl}}\|_2} \approx \frac{\|\mathbf{J}_{\epsilon_\theta}(\tilde{z}_t^{\mathrm{pr}})\delta \tilde{z}_t^{\mathrm{pr}}\|_2}{\|\mathbf{J}_{\epsilon_\theta}(z_t^{\mathrm{cl}})\delta z_t^{\mathrm{cl}}\|_2}. \tag{33}$$

Using the spectral norm bounds from (30):

$$\|\Delta z_t^{\mathrm{pr}}\|_2 \propto \sigma_{\max}(\mathbf{J}_{\epsilon_\theta}(\tilde{z}_t^{\mathrm{pr}})) \cdot \|\delta \tilde{z}_t^{\mathrm{pr}}\|_2 > \sigma_{\max}(\mathbf{J}_{\epsilon_\theta}(z_t^{\mathrm{cl}})) \cdot \|\delta z_t^{\mathrm{cl}}\|_2 \propto \|\Delta z_t^{\mathrm{cl}}\|_2. \tag{34}$$

Since $K_{\mathrm{pr}} > K_{\mathrm{cl}}$ and $\|\delta \tilde{z}_t^{\mathrm{pr}}\|_2 > \|\delta z_t^{\mathrm{cl}}\|_2$, it follows strictly that:

$$\|\Delta z_t^{\mathrm{pr}}\|_2 > \|\Delta z_t^{\mathrm{cl}}\|_2. \tag{35}$$

This derivation confirms that DDIM Inversion acts as an inherent semantic magnifier. For clean images, the reconstruction process is a stable identity mapping. For protected images, the latent perturbations interact with the high-gradient regions of the noise predictor, causing the reconstruction error to accumulate and amplify along the deterministic trajectory. This explains the pronounced degradation in visual quality observed specifically in protected samples.

## D. Dataset Descriptions

We conduct experiments on two widely used benchmarks, **WikiArt** and the **Pokémon** dataset, which are commonly adopted in studies on personalization and mimicry protection methods. All images are center-cropped and resized to $512 \times 512$ before training and evaluation.

**WikiArt.** The WikiArt dataset contains artworks from 129 artists with a total of 81,444 paintings, covering a wide range of artistic styles and periods. We focus on artists with more than 500 available artworks to ensure sufficient data diversity. From this pool, we randomly select four artists and treat each artist as an independent concept subset, namely *albrecht-durer*, *camille-pissarro*, *childe-hassam*, and *claude-monet*. For each artist subset, we randomly sample four paintings as the fine-tuning data used for personalization. For evaluation, we further sample 500 paintings from the remaining artworks of the same artist to construct a reference set for computing evaluation metrics such as FID and CLIP score.

**Pokémon.** The Pokémon dataset consists of 833 images depicting various Pokémon characters. We randomly select four characters, namely *pikachu*, *charizard*, *pansage*, and *sawsbuck*, and treat each character as a distinct concept subset. For each subset, four images are randomly selected and used as fine-tuning data. For evaluation, we randomly sample 500 images from the entire Pokémon dataset to construct a reference set for computing evaluation metrics.

## E. Fine-tuning Details

We provide detailed implementation settings for the three personalization methods evaluated in this work: DreamBooth, LoRA fine-tuning, and Textual Inversion. These details complement the high-level descriptions presented in the main paper.

**DreamBooth.** For DreamBooth-based personalization, we fine-tune both the U-Net and the text encoder. The learning rate is set to $5 \times 10^{-7}$, and the model is trained for 2,000 steps with a batch size of 2. For the WikiArt dataset, we use the instance prompt *"a painting in the style of sks"* and the class prompt *"a painting"*. For the Pokémon dataset, the instance prompt is *"a photo of a sks pokemon character"*, and the class prompt is *"a photo of a pokemon character"*. In all cases, the identifier *sks* denotes a unique concept token associated with the target artist or character.

**LoRA Fine-tuning.** For LoRA-based personalization, we add LoRA adapters with a rank of 4 only to the U-Net, while keeping the text encoder frozen. We train the model for 2,000 steps with a learning rate of $5 \times 10^{-5}$ and a batch size of 2. For WikiArt and Pokémon, we use the prompts *"a painting by artist [ARTIST]"* and *"a photo of a [CHARACTER] pokemon character"*, respectively, where *[ARTIST]* and *[CHARACTER]* denote the target artist and character concepts.

**Textual Inversion.** For Textual Inversion, we optimize the embedding of a newly introduced token while keeping the diffusion model parameters fixed. The embedding is trained using a learning rate of $5 \times 10^{-4}$ for 3,000 optimization steps with a batch size of 4. Each learned token is represented by 8 embedding vectors. The initializer token is set to *"painting"* for WikiArt and *"pokemon"* for the Pokémon dataset. The corresponding prompts are *"a painting in the style of S\*"* for WikiArt and *"a photo of a S\* pokemon character"* for Pokémon, where *S\** denotes the learned textual inversion token.

## F. Baseline Descriptions

### F.1. Mimicry Protection Methods

We provide additional implementation details for the mimicry protection baselines evaluated in our experiments. The perturbation budget for all degradation-based protection methods is set to $\delta = 16/255$ by default. For Glaze, since it does not expose explicit budget control in its official application, we adopt the maximum available perturbation strength.

**Glaze.** Glaze is a degradation-based mimicry protection method originally designed to prevent style imitation. It first employs style transfer techniques to generate a stylized version of the original artwork in the target style. It then optimizes minimal perturbations that shift the artwork's representation in the model's feature space toward that of the stylized target, thereby misleading the model into learning an incorrect style during fine-tuning. In our experiments, we use the latest publicly available version, Glaze 2.1, which provides improved robustness compared to earlier releases.

**Mist.** Mist formulates mimicry protection as a fused adversarial optimization problem for diffusion models. Specifically, it generates protective perturbations by maximizing the diffusion loss, which serves as a semantic objective to disrupt the model's ability to correctly interpret the image content. At the same time, Mist minimizes a VAE reconstruction loss by aligning the perturbed image's latent representation with that of a predefined target image, thereby ensuring the robustness and improving the transferability of the protection across different generation settings. Following the original work, perturbations are optimized using PGD with a step size of $1/255$ for 100 iterations in our experiments.

**Anti-DreamBooth.** Anti-DreamBooth explicitly targets the DreamBooth personalization process by simulating fine-tuning dynamics using surrogate models. It proposes two variants, ASPL and FSMG, both of which aim to optimize adversarial perturbations by maximizing the diffusion loss such that the personalized model fails to properly learn the target concept. In our experiments, we adopt the ASPL variant, which alternates between optimizing model parameters and perturbations. Perturbations are optimized using PGD with a step size of $5 \times 10^{-3}$ for 50 iterations, following the original configuration.

**MetaCloak.** MetaCloak addresses the limited transferability of adversarial protective perturbations by formulating mimicry protection as a meta-learning problem. The method constructs an ensemble of surrogate models corresponding to different fine-tuning stages and optimizes perturbations that generalize across these models. To further enhance robustness, the protected images are subjected to data augmentations including Gaussian blur, horizontal flipping, and center cropping during optimization. Following the original work, we use five surrogate models and optimize perturbations with PGD using a step size of $1/255$ for a total of 4,000 iterations.

**DIAGNOSIS.** DIAGNOSIS is a detection-based mimicry protection method that embeds predefined geometric distortions into protected images. These distortions are designed to be learned during personalization and reproduced in generated outputs, enabling subsequent detection via a dedicated binary classifier. Following the original work, we adopt the unconditional injection setting, where the distortion is applied directly to the images without textual triggers, and set the warping strength to 2.

**SIREN.** SIREN embeds a task-aligned coating into protected images by optimizing the diffusion loss, encouraging the model to learn the embedded signal as a semantically relevant feature. Detection is performed using a hypersphere-based feature extractor to identify the presence of the coating in images. We use the official implementation with the provided encoder and decoder trained on the Pokémon dataset. For evaluation, the numbers of watermarked and clean samples used in the Kolmogorov–Smirnov (KS) test (Massey Jr, 1951) are both set to $n = m = 30$, the significance level is fixed to $\alpha = 0.01$, and the detection process is repeated 10,000 times to obtain stable statistical results.

## F.2. Perturbation Removal Attacks

**Diffusion Attack (DA).** Diffusion Attack removes protective perturbations by reconstructing images through a diffusion process. Specifically, noise is first added to the protected image to disrupt potential perturbations, and the image is then reconstructed via the reverse diffusion process, projecting it back onto the learned data manifold. In our experiments, we implement DA using the Stable Diffusion v2.1-base (SD-v2.1) (AI, 2022) with the number of diffusion steps set to 60.

**VAE Attack (VA).** VAE Attack purifies protected images by reconstructing them through a variational autoencoder. The protected image is first encoded into the latent space and then decoded back to the pixel space, where the reconstruction process suppresses perturbation artifacts. We use the `cheng2020-anchor` (Cheng et al., 2020) model to perform VA, and set the compression factor to 3, where lower values correspond to stronger attack strength.

**IMPRESS.** IMPRESS is motivated by the observation that clean images generally admit lower reconstruction error under a VAE than perturbed images. The method removes protective perturbations by directly optimizing the protected image to minimize the VAE reconstruction loss. Following the original work, we constrain the perturbation budget using LPIPS with a threshold of 0.1 and perform the optimization for 3,000 steps.

**Noisy Upscaling (NU).** Noisy Upscaling first injects Gaussian noise into the protected image, and then restores the image resolution using a diffusion-based upscaling model. This process mitigates perturbation-induced degradation by reconstructing the image at a higher resolution. Following the original work, we add Gaussian noise with a standard deviation of 0.1 and perform upscaling using the `stabilityai/stable-diffusion-x4-upscaler` (Stability AI, 2023) model with the noise level set to 320.

**CAT.** CAT is a model-level perturbation removal attack that targets distortions in the latent space rather than the input image. It introduces LoRA modules into the VAE and trains the encoder and decoder to minimize the reconstruction error of protected images, effectively realigning distorted latent representations to the natural data distribution. We adopt the CAT-both setting, where both the encoder and decoder are fine-tuned with LoRA. The LoRA rank is set to 128, with a learning rate of $1 \times 10^{-4}$, and the model is trained for 1,000 steps, consistent with the original configuration.

*Table 4.* [RQ2] Purification performance on DreamBooth. **Bolded** values denote the best performance; *underlined italicized* values indicate the second best.

| | Glaze | | Mist | | Anti-DreamBooth | | Metacloak | | DIAGNOSIS | | | SIREN | | |
|---|---|---|---|---|---|---|---|---|---|---|---|---|---|---|
| | FID | CS | FID | CS | FID | CS | FID | CS | WACC | FID | CS | WACC | FID | CS |
| None | 245.97 | 0.777 | 315.51 | 0.747 | 325.27 | 0.695 | 280.51 | 0.720 | 100.00% | 144.37 | **0.895** | 96.07% | 130.76 | **0.893** |
| DA | 262.08 | 0.749 | 359.87 | 0.720 | 314.16 | 0.699 | 297.66 | 0.717 | 100.00% | 150.56 | *0.893* | **0.00%** | 133.19 | **0.893** |
| VA | 222.39 | 0.825 | *228.14* | 0.804 | 240.33 | 0.743 | 251.17 | 0.787 | 100.00% | 138.59 | 0.887 | 0.87% | 143.42 | 0.880 |
| IMPRESS | **216.38** | 0.827 | 321.23 | 0.753 | 268.45 | 0.727 | 282.22 | 0.722 | 100.00% | 160.23 | 0.851 | 99.67% | 136.93 | 0.867 |
| NU | 230.06 | **0.869** | 272.48 | **0.842** | 276.40 | *0.803* | 267.57 | *0.811* | 100.00% | 149.06 | 0.877 | 17.31% | 143.76 | 0.870 |
| CAT | 248.90 | 0.782 | 355.88 | 0.747 | 303.07 | 0.685 | 272.97 | 0.726 | 100.00% | *131.09* | 0.889 | 99.95% | **124.77** | *0.889* |
| GrIDPure | 231.35 | 0.805 | 266.44 | 0.786 | 244.31 | 0.739 | *248.32* | 0.782 | 100.00% | 150.55 | 0.882 | 18.80% | 147.60 | 0.886 |
| PDM-Pure | 240.33 | 0.800 | 244.02 | 0.812 | *236.41* | 0.786 | 253.91 | 0.787 | *94.50%* | 143.98 | 0.888 | **0.00%** | 137.90 | **0.893** |
| DIRP (Ours) | *217.87* | *0.840* | **222.54** | *0.817* | **205.95** | **0.817** | **212.71** | **0.828** | **30.80%** | **127.46** | 0.880 | *0.36%* | *129.00* | 0.878 |

*Table 5.* [RQ2] Purification performance on LoRA. **Bolded** values denote the best performance; *underlined italicized* values indicate the second best.

| | Glaze | | Mist | | Anti-DreamBooth | | Metacloak | | DIAGNOSIS | | | SIREN | | |
|---|---|---|---|---|---|---|---|---|---|---|---|---|---|---|
| | FID | CS | FID | CS | FID | CS | FID | CS | WACC | FID | CS | WACC | FID | CS |
| None | 191.26 | *0.827* | 339.81 | 0.726 | 271.19 | 0.733 | 227.39 | 0.770 | 31.30% | 122.61 | 0.851 | 64.71% | 118.84 | 0.862 |
| DA | 210.66 | 0.813 | 373.06 | 0.694 | 277.68 | 0.720 | 258.68 | 0.750 | 32.00% | 130.63 | 0.849 | 0.91% | 126.90 | 0.864 |
| VA | 201.56 | 0.804 | 229.52 | 0.786 | 240.03 | 0.766 | 221.87 | 0.761 | 14.90% | 123.26 | 0.849 | 2.55% | 125.73 | 0.856 |
| IMPRESS | *181.31* | 0.802 | 349.80 | 0.704 | 263.12 | 0.740 | 288.58 | 0.746 | 21.90% | 121.10 | 0.836 | 1.76% | 124.83 | 0.834 |
| NU | 193.59 | **0.828** | *212.49* | **0.827** | 228.75 | 0.791 | *208.33* | 0.780 | 48.80% | 136.45 | 0.841 | 1.93% | 134.29 | 0.846 |
| CAT | 184.60 | 0.820 | 291.22 | 0.723 | 256.58 | 0.729 | 249.70 | 0.720 | 54.30% | **108.95** | *0.859* | 96.77% | **111.31** | **0.871** |
| GrIDPure | 209.70 | 0.800 | 249.30 | 0.770 | 253.05 | 0.758 | 218.30 | 0.746 | 26.70% | *113.17* | 0.840 | **0.00%** | *116.74* | 0.847 |
| PDM-Pure | 228.63 | 0.790 | 232.18 | 0.792 | *225.95* | *0.802* | 223.08 | *0.788* | *12.70%* | 125.86 | 0.853 | *0.63%* | 130.56 | *0.869* |
| DIRP (Ours) | **178.89** | 0.819 | **207.75** | *0.806* | **188.47** | **0.803** | **185.69** | **0.808** | **0.10%** | 117.08 | **0.863** | 2.88% | 117.75 | 0.855 |

**GrIDPure.** GrIDPure is a diffusion-based purification attack that decomposes the protected image into overlapping local grid patches and applies iterative diffusion denoising to each patch independently. By restricting purification to local regions, the method suppresses perturbation artifacts while preserving the global structure of the image. Following the original work, we set the number of diffusion steps per patch to 10 and run 20 purification iterations with a blending weight of $\gamma = 0.1$.

**PDM-Pure.** PDM-Pure is a purification attack that leverages the cascaded DeepFloyd IF pixel-space diffusion model, motivated by the observation that pixel-space diffusion models possess inherent robustness against adversarial perturbations. The method purifies protected images through a two-stage pipeline: Stage II (`IF-II-L-v1.0`) denoises the image at low resolution, and Stage III (`stable-diffusion-x4-upscaler`) restores high-resolution details. Following the original implementation, we apply 10 resampling steps in Stage II with a guidance scale of 7 and 5 support noise-less q-sample steps, and use 50 sampling steps in Stage III with a guidance scale of 4.0.

**DIRP (Ours).** DIRP optimizes a pixel-space purification residual using Adam with the learning rate of 1.0 for 500 optimization steps, while constraining the perturbation magnitude by an $\ell_\infty$ budget of $\varepsilon = 24/255$. All experiments are conducted on a machine equipped with an Nvidia GeForce RTX 4090 GPU.

# G. DreamBooth and Lora Personalization Experiments

We provide a complementary analysis of DreamBooth and LoRA personalization experiments, extending the discussion of RQ2 in Section 4.2. Quantitative results are reported in Table 4 for DreamBooth and Table 5 for LoRA. Following the same evaluation protocol as RQ2, we analyze the effectiveness of DIRP in removing both degradation-based and detection-based protective perturbations under different fine-tuning paradigms.

### G.1. DreamBooth

As shown in Table 4, DreamBooth fine-tuning without purification causes severe degradation for all degradation-based protection methods, with extremely high FID values and reduced CLIP similarity, indicating that full model fine-tuning

amplifies the impact of protective perturbations. For detection-based protections, DreamBooth reliably memorizes watermark-related features, achieving a detection accuracy of nearly 100% for both DIAGNOSIS and SIREN. In contrast, DIRP consistently delivers the strongest overall performance, substantially improving generation quality and reducing DIAGNOSIS WACC to 30.80% while maintaining competitive image fidelity. These results show that even under the most challenging personalization setting, DIRP effectively removes off-manifold perturbations prior to fine-tuning.

### G.2. LoRA Fine-tuning

The results under LoRA fine-tuning are reported in Table 5. Without purification, degradation-based protections still cause noticeable quality degradation, with Mist reaching an FID of 339.81, while detection-based methods remain effective. In contrast, DIRP consistently achieves the best overall performance, reducing FID to below 200 for degradation-based protections and suppressing detection accuracy to near-zero levels while preserving strong perceptual quality. These results indicate that constraining trainable parameters alone is insufficient to neutralize protective perturbations, whereas purifying the training data effectively prevents LoRA from capturing protective perturbation patterns.

