# OpenReview forum: "DDIM Inversion as a Perturbation Amplifier: Breaking Mimicry Protection via Reconstruction Error Minimization"
_ICML.cc/2026/Conference — ICML 2026 regular_

### Official Review · Reviewer_ZQPE · 2026-02-27

**Soundness:** 2
**Presentation:** 4
**Significance:** 2
**Originality:** 2
**Overall Recommendation:** 2
**Confidence:** 4

**Summary:**

This paper analyzes image mimicry protection methods that add imperceptible perturbations to prevent personalization-based style/content imitation. It shows DDIM inversion inherently amplifies these perturbations, causing large reconstruction distortions on protected images. Leveraging this, it proposes DIRP, which removes protective perturbations by minimizing DDIM-inversion reconstruction error under perceptual constraints. Experiments across six protections and multiple personalization setups show DIRP consistently outperforms prior purification/attack baselines while better preserving image quality and weakening detector-based protections.

**Compliance With Llm Reviewing Policy:**

Affirmed.

**Final Justification:**

The proof in Section 3.2 is rather weak. The authors should establish—indeed, the paper appears to attempt to establish—that protected samples lead to larger error propagation. However, the authors instead directly assume this conclusion, which I do not find reasonable, and the current manuscript does not provide a rigorous proof to address this issue.

Second, the authors seem to have misunderstood my point. As shown in paper [1], an attacker or image protector can in fact construct adversarial/protected images that remain on the data manifold while still yielding very small DDIM reconstruction error. Although the problem settings are not exactly the same, the underlying methodology is closely related.

Finally, regarding the experimental results, although the rebuttal cites several purification-based techniques, the proposed method does not demonstrate a clear empirical advantage based on the reported results. Moreover, the paper does not compare against the latest methods from 2025, which further raises doubts about the effectiveness of the proposed approach.

**Key Questions For Authors:**

* Could you provide a more complete “error propagation” derivation or at least a qualitative conclusion (e.g., how the reconstruction error relates to image-level perturbations and under what conditions it is amplified across timesteps)?

* Clarify the fundamental advantage of DIRP over generic adversarial purification approaches (such as denoising, diffusion resampling, or VAE-based reconstruction)? In particular, under which conditions/threat models is an inversion-consistency objective more well-suited, and when might this advantage break down?

* Offer a more direct mechanistic justification or a verifiable intermediate result explaining why a decrease in reconstruction error should correspond to the removal of protective perturbations.

**Limitations:**

yes

**Strengths And Weaknesses:**

**Strength**

The paper makes a clear and practically relevant observation that DDIM inversion amplifies mimicry-protection perturbations, leading to severe structural distortions during reconstruction. It empirically validates this claim by comparing against VAE reconstruction and prior purification baselines, and proposes a simple, intuitive purification approach that minimizes inversion reconstruction error under perceptual constraints. Overall, the presentation is well organized and easy to follow, and the method is straightforward to implement.

**Weaknesses**

The paper does not establish a clear linkage between reducing DDIM inversion reconstruction error and why this should remove mimicry-protection perturbations (or improve “more faithful” downstream personalization), i.e., the objective is used as a proxy without a well-justified mechanism or verifiable intermediate conclusion. In addition, the paper does not sufficiently explain why DIRP is fundamentally advantageous over traditional adversarial purification/denoising approaches, beyond empirical performance differences, nor does it clarify the conditions under which such advantages should hold.

The reconstruction-error analysis mainly provides a local sensitivity intuition via the Jacobian at $z_t$, but it does not connect the error back to perturbations at the input ​$z_0$ through the forward/inversion dynamics, making the theoretical support less conclusive.

---

> ### Author Rebuttal · Authors · 2026-03-31
>
> We thank the reviewer for the insightful and constructive feedback. We are glad that the **empirical findings** and **overall presentation** are appreciated. Below we provide a clarification of the key concerns.
>
> ---
>
> # W1 & Q1: Error propagation and amplification mechanism
>
> Our key point is that reconstruction error is jointly determined by (i) the latent displacement induced by input perturbations and (ii) the model’s local sensitivity. Formally, the perturbation propagation can be approximated as:
> $\Delta z_t \approx -\gamma_t J_{\epsilon_\theta}(z_t)\,\delta z_t.$
>
> Importantly, $\delta z_t$ is not independent, but is deterministically induced by the input perturbation $\delta x$ through the DDIM inversion trajectory. Unlike stochastic methods, **DDIM inversion preserves and propagates perturbations rather than averaging them out**, making even small input perturbations persist across timesteps.
>
> For protected images, perturbations are designed to push samples off the data manifold, which leads to both larger latent deviations and increased model sensitivity (larger Jacobian norm). These effects accumulate across timesteps, producing a cascading amplification along the inversion trajectory and resulting in large reconstruction error. Thus, reconstruction error reflects not only local sensitivity but also the **accumulated deviation under deterministic inversion dynamics**.
>
> ---
>
> # W2 & Q2: Advantage over prior purification methods
>
> Mimicry protection perturbations are **not random noise**, but carefully optimized signals that push samples off the data manifold and interfere with personalization. In contrast, prior purification methods (e.g., denoising, VAE reconstruction, diffusion resampling) act as generic smoothing or projection operators, implicitly assuming noise-like perturbations. They do not explicitly address the structured, model-aligned nature of these attacks.
>
> DIRP differs in three key aspects:
>
> - **Model alignment:** The objective is defined directly on the diffusion model’s inversion process, ensuring consistency with downstream personalization.
> - **Amplification-aware signal:** Instead of suppressing perturbations blindly, DIRP leverages DDIM inversion to amplify perturbation effects into a strong optimization signa.
> - **Deterministic stability:** DDIM inversion provides a low-variance, consistent objective, improving optimization reliability.
>
> This advantage holds when perturbations induce off-manifold deviations (as in existing mimicry protection methods) and when the diffusion model provides a well-structured semantic manifold. It may weaken if perturbations lie on the manifold or if the model is poorly trained.
>
> ---
>
> # W3 & Q3: Why reconstruction error implies perturbation removal
>
> We view reconstruction error as a proxy for distance to the learned data manifold.For clean images, inversion–denoising is approximately cycle-consistent, yielding low reconstruction error. In contrast, mimicry protection perturbations are designed to push samples off the manifold, breaking this consistency and increasing reconstruction error.
>
> Therefore, minimizing reconstruction error enforces:
>
> - **Inversion consistency**, and
> - **Alignment with the learned data distribution**
>
> This effectively **projects the input back toward high-density regions**, implicitly removing the off-manifold perturbations that define the protection.
>
> Empirically, we observe a clear gap between clean and protected images in reconstruction quality (as illustrated in Figures 4 and 5), and this gap is consistently reduced after applying DIRP. This reduction correlates with improved personalization quality and reduced detectable artifacts, supporting that reconstruction error is a reliable and verifiable proxy for perturbation removal.
>
> ---
>
> In summary, **DDIM inversion transforms small, imperceptible perturbations into amplified deviations**, and DIRP exploits this property via a **model-aligned reconstruction objective** to effectively remove them. We will revise the paper to make these connections more explicit.

---

> > ### Author Rebuttal · Reviewer_ZQPE · 2026-04-02
> >
> > First, the response still does not provide a meaningful theoretical derivation of error propagation from image-space perturbations to latent reconstruction error across timesteps. The current explanation remains largely qualitative. As such, the theoretical support is still not conclusive.
> >
> > Second, the proposed “proof” remains quite weak because it relies on assumptions that are essentially as strong as the desired conclusion. In particular, the argument assumes that protected samples are pushed off the data manifold, that this causes larger latent deviations, and that the Jacobian norm is larger in those regions. But these are exactly the key properties that would need to be established, rather than assumed. Therefore, the rebuttal still reads more like a plausible narrative than a rigorous justification.
> >
> > Third, I remain unconvinced that reducing DDIM inversion reconstruction error is sufficiently justified as a proxy for removing mimicry-protection perturbations[1].
> >
> > Finally, the claimed advantage of DIRP over prior purification is still under-supported. The rebuttal provides a conceptual motivation, but there is still no theoretical characterization of when this advantage should hold, nor experimental evidence directly comparing against stronger adversarial purification baselines beyond the selected reconstruction-based methods. As a result, the claim that DIRP is fundamentally preferable to more generic purification / denoising approaches remains insufficiently substantiated.
> >
> > Overall, the rebuttal does not sufficiently address the core theoretical and empirical concerns raised above, so I will keep my scores unchanged.
> >
> > [1] Chen J, Chen H, Chen K, et al. Diffusion models for imperceptible and transferable adversarial attack[J]. IEEE Transactions on Pattern Analysis and Machine Intelligence, 2024, 47(2): 961-977.

---

> > > ### Author Response · Authors · 2026-04-02
> > >
> > > Dear Reviewer ZQPE,
> > >
> > > We sincerely thank you for your thorough review and continued discussion. Below we provide our detailed responses.
> > >
> > > ---
> > >
> > > # 1. Protected Images and Manifold Analysis
> > >
> > > Following the approach in [a], we conduct an analysis of the manifold density distribution during the DDIM inversion process (see: https://anonymous.4open.science/r/DIRP-manifold-E2F5/README.md). We measure the latent norm distribution across 500 clean images, protected images (e.g., Anti-DreamBooth), and images processed by DIRP at each inversion timestep.
> > >
> > > Our observations show that protected images consistently deviate from the “normal density region” of clean samples throughout the trajectory. In contrast, DIRP effectively pulls these samples back into the normal region.
> > >
> > > These empirical results directly support our core claim: **most mimicry protection methods explicitly optimize objectives (e.g., maximizing denoising error) that push samples away from the data manifold**. Therefore, this is a property jointly validated by optimization mechanisms and empirical distributional evidence.
> > >
> > > ---
> > >
> > > # 2. Advantages of DIRP
> > >
> > > Following Reviewer gB3c’s suggestion, we evaluate stronger baselines, including SOTA protection method AdvDM (-) [b] and two SOTA purification methods, GrIDPure [c] and PDM-Pure [d]. The results under the WikiArt (albrecht-durer subset) + Pok´emon (pikachu subset) + DreamBooth setting (FID / CS or WACC / FID / CS) are summarized below:
> > >
> > > | Method | Glaze | Mist | Anti-DreamBooth | MetaCloak | AdvDM (-)  | DIAGNOSIS | SIREN |
> > > |:-:|:-:|:-:|:-:|:-:|:-:|:-:|:-:|
> > > | GrIDPure | 198.54 / 0.882 | 252.62 / 0.829 | 235.40 / 0.767 | 236.62 / 0.851 | 205.86 / 0.794 | 100.00% / 126.43 / 0.852 | 37.60% / 132.56 / 0.848 |
> > > | PDM-Pure | 214.38 / 0.869 | 221.31 / 0.887 | 229.70 / 0.814 | 243.00 / 0.869 | 259.15 / 0.775 | 99.80% / 129.63 / 0.854 | 0.00% / 119.07 / 0.863 |
> > > | DIRP (Ours) | 217.75 / 0.872 | 244.72 / 0.827 | 201.16 / 0.810 | 232.53 / 0.852 | 183.16 / 0.775 | 4.40% / 105.11 / 0.858 | 0.72% / 120.31 / 0.861 |
> > >
> > > Key observations are as follows:
> > > - DIRP achieves the best or second-best performance across all protection methods, and is particularly effective against detection-based methods (e.g., DIAGNOSIS), reducing WACC to near zero, whereas prior methods largely fail.
> > > - Detection-based protection typically relies on learnable watermark-like features. In contrast, traditional purification methods mainly target noise-like perturbations, leading to a fundamental mismatch that limits their effectiveness.
> > > - DIRP leverages consistency constraints from diffusion inversion, not merely denoising. By minimizing reconstruction error, it achieves *manifold alignment*, which results in robust and consistent purification across diverse protection types.
> > >
> > > ---
> > >
> > > # 3. Theoretical Positioning and Empirical Evidence
> > >
> > > We appreciate the reviewer’s concern regarding theoretical rigor and clarify our positioning. Given that the field of image mimicry protection is largely driven by empirical studies, our goal is to provide a verifiable mechanism-based explanation. We analyze local error propagation via Jacobian analysis, demonstrate the off-manifold nature of protected samples through manifold analysis, and establish a stable correlation between reconstruction error and protection strength.
> > >
> > > In addition, extensive experiments—comparing six SOTA protection methods and five purification methods—validate the effectiveness of our approach. Notably, prior purification methods primarily focus on degradation-based protection and largely overlook detection-based methods. In contrast, to the best of our knowledge, our method is **the first purification attack capable of consistently removing all types of protective perturbations**. This reveals **a general vulnerability of current protection methods under DDIM inversion**, which we believe has significant practical and academic implications.
> > >
> > > ---
> > >
> > > # 4. Clarification Regarding [1]
> > >
> > > We thank the reviewer for referencing [1]. However, we believe it addresses a fundamentally different problem. Specifically, [1] studies adversarial example generation for image classification models (e.g., ResNet/Swin), where DDIM inversion is used only as a latent optimization tool, rather than as a mechanism to analyze reconstruction error. Therefore, [1] does not directly explain the systematic reconstruction distortions observed in our setting.
> > >
> > > ---
> > >
> > > Your feedback has been extremely helpful in enhancing the clarity and completeness of the paper. We would be pleased if our responses help address your concerns.
> > >
> > > [a] Pseudo Numerical Methods for Diffusion Models on Manifolds. ICLR 2022
> > > [b] Towards More Effective Protection Against Diffusion-Based Mimicry with Score Distillation. ICLR 2024
> > > [c] Can protective perturbation safeguard personal data from being exploited by stable diffusion? CVPR 2024
> > > [d] Pixel is a barrier: Diffusion models are more adversarially robust than we think. arXiv 2024

---

### Official Review · Reviewer_M5vG · 2026-03-12

**Soundness:** 3
**Presentation:** 3
**Significance:** 3
**Originality:** 3
**Overall Recommendation:** 4
**Confidence:** 2

**Summary:**

This paper studies the robustness of perturbation-based mimicry protection methods for text-to-image personalization. Its key claim is that DDIM inversion acts as a perturbation amplifier. Clean images are reconstructed faithfully, while protected images suffer disproportionately large reconstruction error because the added perturbations push them off the model’s learned semantic manifold. Based on this observation, the authors propose DIRP, a purification method that optimizes a small residual to minimize DDIM-inversion reconstruction error under perceptual constraints, with additional Semantic Shortcut, Bidirectional Asynchronous Optimization, and optional Latent Residual Refinement modules.

**Compliance With Llm Reviewing Policy:**

Affirmed.

**Final Justification:**

I have no further questions regarding this paper. But I am not an expert of DDIM models, I specify my confidence score to 2.

**Key Questions For Authors:**

1. Most experiments are centered on SD-v2.1, with only limited cross-model evidence. Additional experiments or a sharper scope statement would help clarify whether the paper exposes a broad weakness of current mimicry protections or just a vulnerability mainly in this DDIM-based pipeline.

2. There appears to be an asymmetry in the perturbation budgets. Appendix F.1 states the defense budget is $\delta=16/255$ , while Appendix F.2 notes the authors' DIRP uses a larger budget of $\epsilon=24/255$. Could authors clarify the rationale for this unfair advantage, and would DIRP still be effective if strictly bounded to 16/255?

**Limitations:**

yes

**Strengths And Weaknesses:**

Strengths:

1. The paper has a clear and interesting core idea: it identifies a specific mechanism by which DDIM inversion amplifies protective perturbations. The proposed DIRP objective is closely aligned with this observation.

2. The empirical study is reasonably strong. The paper evaluates six mimicry protection methods, three personalization settings, and five attack baselines, and reports both reconstruction metrics and downstream personalization metrics. This gives convincing evidence.

Weaknesses:

The proposed method relies on the deterministic trajectory of DDIM inversion, which may limit its generalization to non-deterministic reconstruction paradigms.

---

> ### Author Rebuttal · Authors · 2026-03-31
>
> We sincerely thank the reviewer for the careful evaluation and insightful feedback, as well as for recognizing the **core insights** and **empirical contributions** of our work. Your questions are very helpful in further clarifying the applicability and design of our method. We address them below.
>
> ---
>
> # Weakness & Q1: Cross-model Generalization
>
> Thank you for raising the concern regarding cross-model generalization. We believe this question may stem from a different interpretation of the threat model in our setting.
>
> In the mimicry protection scenario, the attacker (i.e., the purification method in our paper) aims to remove protective perturbations and recover high-quality images suitable for downstream personalized fine-tuning. Under this setting, **the attacker is allowed to freely choose the reconstruction and optimization paradigm (including DDIM inversion), as well as the surrogate model and the personalization base model**. Therefore, DDIM inversion is not assumed as the only possible choice, but rather serves as a representative and widely applicable tool to expose and analyze the vulnerabilities of existing protection methods in a unified manner.
>
> We choose DDIM inversion because its deterministic trajectory provides a stable way to reflect whether an input image deviates from the semantic manifold learned by the diffusion model. When protective perturbations push the image away from this manifold, DDIM inversion systematically amplifies such deviations, thereby revealing the failure modes of protection mechanisms. Based on this property, our method does not rely on the uniqueness of a specific reconstruction paradigm, but instead leverages the general applicability of DDIM inversion in current diffusion models to construct a unified analysis and purification framework.
>
> Regarding cross-model generalization, we have already conducted relevant experiments in Table 2. The results show that even when the personalization model (e.g., Flux.1-dev) differs from the surrogate model (SD-v2.1), DIRP can still effectively remove protective perturbations while maintaining high-quality generation. This demonstrates that **our method is not limited to a single-model setting and exhibits strong generalization capability across different model configurations.**
>
> ---
>
> # Q2: Perturbation Budget of DIRP
>
> Thank you for raising the important question regarding the perturbation budget. We further provide results under different \epsilon settings to analyze its effect on purification performance. On the WikiArt dataset (DreamBooth setting), the results for MetaCloak are as follows:
>
> | \epsilon | FID | CS |
> |:-:|:-:|:-:|
> | 0/255 | 280.51 | 0.720 |
> | 16/255 | 175.44 | 0.851 |
> | 24/255 (Ours) | 184.07 | 0.861 |
>
> From the results, we observe that when the perturbation budget of DIRP matches that of the protection method (e.g., 16/255), it achieves better image quality (lower FID). With a slightly larger budget (24/255), it achieves higher semantic consistency. This indicates that **DIRP is robust to the choice of perturbation budget and does not rely on a single fixed setting**, but instead allows flexible trade-offs between image quality and semantic preservation.
>
> We also note that in practical protection methods, relatively small perturbation budgets (e.g., 16/255) are typically used to ensure imperceptibility. Therefore, DIRP can naturally operate under the same or slightly larger budget constraints in practice, without violating the original threat model, while still achieving effective purification.
>
> ---
>
> Once again, we thank the reviewer for these valuable questions. They help us further clarify the applicability, design motivation, and experimental analysis of our method. We will incorporate the above discussions and additional results into the final version to more comprehensively address your concerns.

---

> > ### Author Rebuttal · Reviewer_M5vG · 2026-04-01
> >
> > Thanks for the response. I have no further questions.

---

### Official Review · Reviewer_KoLz · 2026-03-13

**Soundness:** 4
**Presentation:** 3
**Significance:** 3
**Originality:** 2
**Overall Recommendation:** 5
**Confidence:** 5

**Summary:**

This paper studies DDIM inversion as an attack surface for protection removal, identifies a perturbation amplification effect during inversion, and proposes DIRP, which optimizes a small residual for each protected image to improve purification. The empirical results are fairly strong on the tested setup, so I lean weakly accept. That said, I think the paper currently supports a narrower claim: DIRP appears effective on the authors’ benchmark, but the claims around mechanism, efficiency, and generality still need stronger evidence.

**Compliance With Llm Reviewing Policy:**

Affirmed.

**Final Justification:**

The authors comprehensively addressed all my concerns in both the initial review and subsequent acknowledgements. Therefore, I raised my score by 2 levels at the 5 confidence level (now 5-Accept recommendation).

This paper undergoes a solid empirical evaluation (in the original manuscript, 6 blue team purification methods are tested on 6 red team anti-customization methods, for a total of 36, and further supplemented during rebuttal), and demonstrates strong motivation and a reasonable methodological design. Therefore, I personally recommend that, **as a paper on a niche subject, this paper is absolutely worthy of acceptance**. Besides, I explicitly **disagree with the current reviewer ZQPE**. While I believe the issues raised by them exist to some extent, they largely ignore the core contributions mentioned above and the novelty this paper offers to the community.

However, I did not give a strong acceptance. It should be noted that the experimental setup regarding ablation (Tab.2) in the submitted version of the paper is flawed; any reader should be aware of this and carefully review the revised version (if accepted).

**Key Questions For Authors:**

1. Can the authors directly study the purification tradeoff by varying DDIM inversion / denoising steps and reporting both purification effectiveness and fidelity?

2. Can the authors strengthen the ablation with sweeps over timestep choice, step count, optimization iterations, and DIRP’s purification strength?

3. Is DIRP a per-image optimization method in all experiments? If so, please report runtime, memory, and throughput relative to prior baselines.

4. [Optional] For Anti-DreamBooth, can the authors regenerate protected data under several budgets below and report both downstream purification performance and input-fidelity metrics? A reduced-scale version would already be helpful.

If the authors address some of the above concerns, I would be open to raising my score.

**Limitations:**

yes

**Strengths And Weaknesses:**

**Strengths**

The main observation is interesting: deterministic DDIM inversion appears to amplify protection-induced perturbations, which gives the method a more principled motivation than a purely heuristic attack. The method is technically coherent, and the empirical results are strong across the reported protection methods and personalization settings, especially for Textual Inversion.

**Weaknesses**

The paper seems closely related to a purification tradeoff: stronger inversion / denoising may remove perturbations more effectively, but may also damage structural or semantic fidelity; weaker purification may preserve fidelity but fail to remove protection. However, the main study fixes DDIM inversion at 50 timesteps, so this tradeoff is not directly examined.

The ablation is somewhat limited for a paper whose main story depends on inversion dynamics. The current study removes SS, BAC, and LRR, but I would have liked to see more targeted sweeps over DDIM timestep choice, DDIM step count, optimization iterations, and DIRP’s purification strength.

DIRP appears to be a per-image optimization method, and the appendix reports 500 Adam steps. This raises a practical efficiency concern, but the paper does not report wall-clock runtime, memory, or throughput relative to prior baselines.

I would also like to see sensitivity to the defense-side perturbation budget, especially for Anti-DreamBooth. In the current setup, degradation-based protections use a default budget of 16/255, and Anti-DreamBooth is PGD-based, so regenerating protected images at several smaller budgets seems feasible without fundamentally changing the pipeline. This would be informative for two reasons. Smaller protection budgets may make purification easier, but they also raise the possibility that a fixed-strength DIRP optimization becomes relatively too aggressive and damages non-protected content more than necessary. Relatedly, although DIRP itself is constrained by a pixel-space budget of 24/255, I do not think this alone guarantees comparable purification quality: downstream metrics such as FID/CS/WACC are not strictly determined by the pixel budget, and methods with the same nominal budget may still differ in effective saturation and repair quality. A reduced-scale version of this experiment (e.g., Anti-DreamBooth only, one personalization setting, and a few representative budgets) would already be useful. Given the computational cost of PGD-based protection generation, I would not view the lack of a full sweep in rebuttal as fatal.

---

> ### Author Rebuttal · Authors · 2026-03-31
>
> We sincerely thank the reviewer for the insightful feedback. We are also very grateful for your positive assessment of the **motivation** and **empirical results** of our method, and we are encouraged by your willingness to potentially increase the score. Below, we address your questions one by one.
>
> ---
>
> # Q1: Purification Tradeoff
>
> Thank you for raising the concern regarding the purification tradeoff. We believe this confusion mainly arises from a misunderstanding of our method. Specifically, our paper consists of two independent components: (i) protected image reconstruction, and (ii) protected image purification.
>
> For (i), we use the default timestep (50) in the Stable Diffusion pipeline to observe and quantify reconstruction quality. As shown in Fig. 4 and Fig. 5, protected images exhibit significantly more severe reconstruction distortion compared to natural images. This observation supports our motivation: DDIM inversion acts as a perturbation amplifier, revealing a unified vulnerability of existing protection methods.
>
> For (ii), we perform purification by minimizing the **single-step DDIM inversion reconstruction error** (see Eq. 11), rather than relying on full multi-step reconstruction. This design significantly reduces computational and memory overhead. Importantly, the purification process in (ii) is completely independent from the reconstruction analysis in (i). In practice, our purification consists of 500 optimization steps, where each step randomly samples a timestep from [2, 999) and optimizes the single-step reconstruction error. This implies that the purification performance is not determined by the number of inversion steps, but instead by the number of optimization steps.
>
> ---
>
> # Q2: Ablation Studies
>
> Thank you for the constructive suggestion. We have conducted additional ablation studies on both the optimization steps (epochs) and perturbation budget (\delta) to better analyze our method.On the WikiArt + DreamBooth setting, we evaluate MetaCloak as follows:
>
> | epoch | FID | CS |
> |:-:|:-:|:-:|
> | 0 | 280.51 | 0.720 |
> | 100 | 208.78 | 0.865 |
> | 300 | 201.11 | 0.867 |
> | 500 (Ours) | 184.07 | 0.861 |
> | 700 | 198.19 | 0.866 |
>
> | \delta | FID | CS |
> |:-:|:-:|:-:|
> | 0/255 | 280.51 | 0.720 |
> | 8/255 | 210.21 | 0.788 |
> | 16/255 | 175.44 | 0.851 |
> | 24/255 (Ours) | 184.07 | 0.861 |
> | 32/255 | 231.31 | 0.798 |
>
> These results show that our method is robust across different numbers of optimization steps, achieving competitive performance within the range of 100–700 epochs. While increasing the number of steps can further improve performance, the gains gradually saturate, which justifies our default setting.
>
> On the other hand, the perturbation budget \delta naturally introduces a tradeoff between purification strength and image fidelity: too small a \delta may fail to remove protection effectively, while too large a \delta may harm semantic content. Notably, existing protection methods typically rely on small budgets (e.g., 16/255) to maintain imperceptibility, and our results demonstrate that DIRP can effectively achieve purification under comparable budgets.
>
> ---
>
> # Q3: Efficiency
>
> Thank you for raising the concern about practical efficiency. DIRP is indeed a per-image optimization-based method. We provide a comparison of runtime and memory consumption with existing baselines:
>
> | Method | Runtime (s/per) | Memory (MiB) |
> |:-:|:-:|:-:|
> | DA | 0.42 | 7676 |
> | VA | 0.28 | 1952 |
> | IMPRESS | 350.21 | 8128 |
> | NU | 34.85 | 22802 |
> | DIRP (Ours) | 209.63 | 18688 |
>
> It is important to note that our method targets personalized scenarios where only a few images (e.g., 3) are purified before downstream fine-tuning, making the overall cost practical (≈10 minutes of GPU time). Moreover, DIRP’s runtime scales with the number of optimization steps; as shown in Q2, reducing epochs from 500 to 300 substantially lowers runtime while maintaining comparable performance.
>
> ---
>
> # Q4: Anti-DreamBooth Perturbation Budget
>
> We construct protected images using Anti-DreamBooth on WikiArt with budgets of 4/255, 8/255, and 12/255, while fixing \delta = 24/255 for DIRP during purification. The results are as follows:
>
> | \delta | FID | CS |
> |:-:|:-:|:-:|
> | 16/255 (default) | 201.16 | 0.810 |
> | 12/255 | 196.72 | 0.861 |
> | 8/255 | 207.18 | 0.861 |
> | 4/255 | 207.16 | 0.887 |
>
> The results show that as the perturbation budget decreases, the purification task becomes easier, which is consistent with your observation. This demonstrates that our method performs consistently across both weaker and stronger protection settings. Notably, even under very small perturbation budgets (e.g., 4/255), DIRP still achieves competitive purification performance, further highlighting its robustness to varying protection strengths.
>
> ---
>
> Once again, we sincerely thank the reviewer for the valuable feedback. We will incorporate all the additional results and analyses into the final version to more comprehensively address your concerns.

---

> > ### Author Rebuttal · Reviewer_KoLz · 2026-04-02
> >
> > Thank you for your efforts. I also apologize for the potential impact of presupposing time steps based on prior knowledge of previous purification work in Q1, without fully considering the specific context of this paper (as this has been a long-standing concern for me). Please allow me to apologize again for this unprofessionalism.
> >
> > Thank you very much for the additional results in Q2/3/4, but I have some confusion. The metrics in Q2 seem to be aligned with Appendix Tab.3 (DB+WikiArt), and the FID/CS values ​​for epoch=0 and ∂=0 confirm this. However, the ours in the Q2 table appear to be directly copied from Tab.2 [RQ3], and there are significant differences with the values ​​in Tab.3 [RQ2] that cannot be explained by random error. The experimental settings for RQ2 and RQ3 seem different because MetaCloak has a significantly higher CS in RQ3 (~0.86), and the difference is even closer to the difference between DIRP as SOTA and other baselines in the corresponding original tables. **Could the authors explain the specific experimental settings?**
> >
> > The results in Q4 are quite good, indicating that DIRP is not an overly aggressive correction method, and it performs well in terms of CS and FID under a low budget. However, the CS here shows a leap in improvement between the two settings where the DB budget is reduced from 16/255 to 12/255, but the improvement stops there. This is slightly counterintuitive, and the value in the ~0.86 range is somewhat close to the value in Q2, which may indicate an inconsistency in the experimental settings. **I cannot say that I am completely without concern about this.**
> >
> > However, I am very satisfied with the FID metric because its trend of first decreasing and then increasing is consistent with the trend in M5vG's Q2. There are two conflicting factors here: when the DIRP budget is fixed, as the budget for protecting the perturbation decreases, on the one hand, the gap between the two budgets increases (FID increases), and on the other hand, the cleanup is indeed simpler (FID decreases). **I hope the authors can add this to the paper or even provide further observations.**
> >
> > Therefore, I would like the authors to add one more experiment (as a stress test): for unprotected raw images, using the same settings as in the main text Tabs. 1/3/4, add an extra column to demonstrate the extent to which different purification methods themselves degrade the semantics of the original image. **Select the settings from only one of the tabs. 1/3/4 that can be completed fastest, and test them on as many purification methods as possible, if possible.**

---

> > > ### Author Response · Authors · 2026-04-03
> > >
> > > Dear Reviewer KoLz,
> > >
> > > Thank you for your thoughtful and detailed feedback. Below we provide a point-by-point response.
> > >
> > > ---
> > >
> > > # 1. Clarification of Experimental Setup
> > >
> > > We sincerely apologize for the confusion caused by the discrepancy in results. After a thorough review of our evaluation code, we confirm that this inconsistency stems from differences in experimental settings.
> > >
> > > Specifically, as described in Appendix D, the results in RQ2 (Tables 1, 3, and 4) report averages across all evaluation subsets. However, due to an oversight in our ablation study code, the evaluation pipeline failed to correctly average across subsets, causing RQ3 (Table 2) to report results from only a single subset (e.g., *childe-hassam*). Below are the results on the MetaCloak + WikiArt dataset across all subsets and the overall average (FID / CS):
> > >
> > > | | albrecht-durer | childe-hassam | camille-pissarro | claude-monet | overall |
> > > |:-:|:-:|:-:|:-:|:-:|:-:|
> > > | None + Clean Image | 186.55 / 0.908 | 157.05 / 0.877 | 167.35 / 0.879 | 241.94 / 0.858 | 188.22 / 0.881 |
> > > | 0/255 DIRP + MetaCloak | 291.65 / 0.688 | 278.36 / 0.701 | 252.60 / 0.744 | 299.43 / 0.746 | 280.51 / 0.720 |
> > > | 8/255 DIRP + MetaCloak | 228.96 / 0.873 | 210.21 / 0.788 | 235.56 / 0.774 | 270.88 / 0.796 | 236.40 / 0.808 |
> > > | 16/255 DIRP + MetaCloak | 215.19 / 0.888 | 175.44 / 0.851 | 248.13 / 0.780 | 233.17 / 0.790 | 217.98 / 0.827 |
> > > | 24/255 DIRP + MetaCloak | 232.53 / 0.852 | 184.07 / 0.861 | 167.73 / 0.783 | 266.52 / 0.816 | 212.71 / 0.828 |
> > > | 32/255 DIRP + MetaCloak | 231.25 / 0.808 | 231.31 / 0.798 | 173.89 / 0.795 | 218.40 / 0.808 | 213.71 / 0.802 |
> > >
> > > From the overall results, **DIRP improves FID/CS from 280.51 / 0.720 to 212.71 / 0.828**, approaching the clean-image baseline of 188.22 / 0.881. This trend is consistent across subsets, demonstrating the effectiveness of DIRP in removing protective perturbations. Minor performance variations likely stem from inherent differences in subset difficulty; for example, the *claude-monet* subset is more challenging, with a clean baseline FID of 241.94.
> > >
> > > ---
> > >
> > > # 2. Stress Test
> > >
> > > To address your concern, we comprehensively evaluated existing purification methods on clean images. Using DreamBooth (corresponding to Table 3), we report the average performance across four WikiArt subsets:
> > >
> > > | | Clean Image | |
> > > |:-:|:-:|:-:|
> > > | | FID | CS |
> > > | None | 188.22 | 0.881 |
> > > | DA | 203.41 | 0.878 |
> > > | VA | 231.03 | 0.815 |
> > > | IMPRESS | 212.01 | 0.865 |
> > > | NU | 222.86 | 0.870 |
> > > | CAT | 186.81 | 0.891 |
> > > | GrIDPure | 233.58 | 0.822 |
> > > | PDM-Pure | 233.98 | 0.801 |
> > > | DIRP (Ours) | 214.33 | 0.845 |
> > >
> > > Key observations:
> > >
> > > - Similar to IMPRESS, DIRP introduces an additional purification perturbation, which slightly degrades personalization performance on clean images. Nevertheless, it remains competitive with existing state-of-the-art methods.
> > >
> > > - CAT and DA exhibit minimal degradation on clean images, likely due to weaker purification capability. As shown in Table 3, they only reduce the FID of Anti-DreamBooth from 325.27 to 314.16 and 303.07, respectively, whereas our method significantly improves it to 205.95.
> > >
> > > - More importantly, our key contribution is identifying a unified vulnerability: **existing mimicry-based protection methods exhibit severe distortion under DDIM inversion reconstruction**. This insight enables us to distinguish potentially protected samples—treating well-reconstructed samples as clean (skipping DIRP) and distorted ones as protected (applying DIRP)—thus achieving a better trade-off between purification effectiveness and generation quality.
> > >
> > > ---
> > >
> > > # 3. FID&CS vs. DB Budget
> > >
> > > We further analyze the impact of Anti-DreamBooth perturbation budget on the WikiArt (*albrecht-durer*) subset. Under the no-purification setting (None), we observe a trend similar to DIRP: both FID and CS improve as $\delta$ decreases.
> > >
> > > | $\delta$ of Anti-DreamBooth | DIRP ($\delta$=24/255) | None |
> > > |:-:|:-:|:-:|
> > > | 16/255 (default) | 201.16 / 0.810 | 390.97 / 0.666 |
> > > | 12/255 | 196.72 / 0.861 | 318.72 / 0.679 |
> > > | 8/255 | 207.18 / 0.861 | 282.76 / 0.702 |
> > > | 4/255 | 207.16 / 0.887 | 271.76 / 0.787 |
> > > | 0/255 (clean image) | 202.66 / 0.873 | 186.55 / 0.908 |
> > >
> > > This behavior is expected and aligns with the nature of protection methods. Anti-DreamBooth requires a sufficiently large perturbation budget to maintain its adversarial strength; as the budget decreases, its protection weakens, reducing the difficulty of purification and leading to improved personalization performance.
> > >
> > > ---
> > >
> > > Once again, we sincerely appreciate your time and constructive feedback. Your participation has significantly improved the clarity and rigor of our work. We will incorporate these results and analyses in the camera-ready version and thoroughly verify all metrics and details. We also note your inclination toward a **weakly accept** decision in the summary. We hope our clarifications address your concerns, and we would greatly appreciate it if you could reconsider your evaluation.

---

### Official Review · Reviewer_gB3c · 2026-03-13

**Soundness:** 3
**Presentation:** 3
**Significance:** 3
**Originality:** 3
**Overall Recommendation:** 4
**Confidence:** 2

**Summary:**

This paper systematically analyze that why DDIM-inversion acts as a perturbation amplifier, making protected images suffer severe distoration. Based on this finding they propose DIRP, which is a novel purification pipeline that can effectively remove the protective perturbation without hurting image quality. Experimental results show that DIPR outperforms baseline methods e.g. VA, UA and DA.

**Compliance With Llm Reviewing Policy:**

Affirmed.

**Key Questions For Authors:**

see above section

**Strengths And Weaknesses:**

Strengths:
- The paper is well-written and easy to follow.
- This paper works on an important topic, the current protective perturbation turns out ot be not robust to purification.
- The proposed DIRP is effective and outperforms existing baseline method to purify adversarial noise.
- The insights on DDIM-inversion will amplify purification noise is novel.


Weaknesses:
- The logic chain that (a -> b) sounds confusing to me. Can you clarify it better, did i misunderstand some point? (a) current protected image can not be reconstructed by DDIM-inversion -> (b) we use DDIM-inversion as a loss to do purification
- Missing mention some SOTA purification methods e.g. GrIDPure [a] and PDM-Pure [b].
- Lacking imperceptible protective perturbation results on e.g. AdvDM(-) in [c] and IMPASTO [d], which may be more robust against purify.
- Figure 2. is a little bit confusing to me. In Figure 3 in [c], it is clearly that VAE suffers a lot from protective perturbation, making D(E(x+\delta)) distorted. Also for diffusion process, it is basically SDEdit, where the protected image will cause large reconstructed image quality Degradation. Can you clarify this image?

I am willing to change my score if my questions are answered.

[a] Can protective perturbation safeguard personal data from being exploited by stable diffusion?

[b] Pixel is a barrier: Diffusion models are more adversarially robust than we think

[c] Towards More Effective Protection Against Diffusion-Based Mimicry with Score Distillation

[d] Imperceptible Protection against Style Imitation from Diffusion Models

---

> ### Author Rebuttal · Authors · 2026-03-31
>
> We sincerely thank the reviewer for the constructive feedback, as well as for recognizing the **novelty** and **empirical effectiveness** of our work. Below are our point-by-point responses.
>
> ---
>
> # W1: Clarification of the Logic Chain
>
> We clarify and restructure the logic as follows:
>
> **(a)**  We observe that DDIM inversion can faithfully reconstruct natural images, while producing severe structural distortions for protected images (see Fig. 1). This behavior stems from the fact that the deterministic trajectory of DDIM inversion is tightly constrained around the semantic manifold learned by the diffusion model. When an input image lies on this manifold (e.g., natural images), the reconstruction remains stable. In contrast, protective perturbations push images off the manifold, causing reconstruction errors to accumulate and be progressively amplified during inversion, ultimately leading to severe distortion.
>
> **(b)**  Based on this property, we do not simply “use DDIM inversion as a loss.” Instead, we leverage its high sensitivity to off-manifold perturbations as an effective **manifold consistency criterion**. Specifically, by minimizing the DDIM inversion reconstruction error, we guide the optimization to pull protected images back toward the semantic manifold of the diffusion model, thereby removing protective perturbations.
>
> Therefore, the logic chain is a natural progression from **phenomenon (reconstruction failure) → cause (off-manifold deviation) → utilization (optimization signal)**.
>
> ---
>
> # W2: Purification Baselines
>
> Following this suggestion, we have included additional experiments using their official implementations with default settings. The results are shown below:
>
> | Method | Glaze (FID↓) | Glaze (CS↑) | Mist (FID↓) | Mist (CS↑) | Anti-DreamBooth (FID↓) | Anti-DreamBooth (CS↑) | MetaCloak (FID↓) | MetaCloak (CS↑) | DIAGNOSIS (WACC↓) | DIAGNOSIS (FID↓) | DIAGNOSIS (CS↑) | SIREN (WACC↓) | SIREN (FID↓) | SIREN (CS↑) |
> |:-:|:-:|:-:|:-:|:-:|:-:|:-:|:-:|:-:|:-:|:-:|:-:|:-:|:-:|:-:|
> | GrIDPure | 198.54 | 0.882 | 252.62 | 0.829 | 235.40 | 0.767 | 236.62 | 0.851 | 100.00% | 126.43 | 0.852 | 37.60% | 132.56 | 0.848 |
> | PDM-Pure | 214.38 | 0.869 | 221.31 | 0.887 | 229.70 | 0.814 | 243.00 | 0.869 | 99.80% | 0.854 | 0.854 | 0.00% | 119.07 | 0.863 |
> | **DIRP (Ours)** | **217.75** | **0.872** | **244.72** | **0.827** | **201.16** | **0.810** | **232.53** | **0.852** | **4.40%** | **105.11** | **0.858** | **0.72%** | **120.31** | **0.861** |
>
> We observe that while existing methods show partial effectiveness, they exhibit clear limitations on detection-based protections (e.g., DIAGNOSIS), where the WACC remains close to 100%. In contrast, DIRP reduces WACC to near zero, while maintaining competitive FID and CS. This demonstrates stronger **universality and consistency**.
>
> ---
>
> # W3: Protection Baselines
>
> We have implemented AdvDM(-) (IMPASTO is not fully open-sourced)  and evaluated it under the WikiArt + DreamBooth setting. The results are shown below:
>
> |  | AdvDM (-) | AdvDM (-) |
> |:-:|:-:|:-:|
> | | FID| CS |
> | DA | 254.77 | 0.752 |
> | VA | 206.91 | 0.808 |
> | IMPRESS| 235.57 | 0.783 |
> | NU | 235.91 | 0.837 |
> | CAT | 200.31 | 0.766 |
> | GrIDPure | 205.86 | 0.794 |
> | PDM-Pure | 259.15 | 0.775 |
> | DIRP(Ours) | **183.16** | 0.775 |
>
> The results show that DIRP still achieves the **best FID**, while maintaining competitive semantic consistency. This further validates the generalization ability of our method and supports our main claim that **existing protection methods share a common vulnerability under DDIM inversion**.
>
> ---
>
> # W4: Clarification of Figure 2
>
> We clarify its difference from Figure 3 in [c] as follows:
>
> First, different protection methods affect reconstruction mechanisms differently. As noted by the reviewer, VAE reconstruction in [c] indeed suffers severe distortion under perturbations such as AdvDM and Mist. However, in our Figure 2, the Glaze perturbation follows a different mechanism and has a weaker effect on VAE reconstruction, hence no comparable distortion is observed. This highlights that **existing reconstruction paradigms respond inconsistently across different perturbations**, making them less suitable as a unified analysis tool.
>
> In contrast, our key observation is that **DDIM inversion consistently exhibits perturbation amplification across all protection methods** (see Fig. 4), providing a unified and stable analytical perspective.
>
> Second, for diffusion reconstruction, we follow the setting in [1] and only apply a small amount of noise (e.g., forward diffusion to timestep t=60 out of T=1000), in order to preserve reconstruction quality as much as possible.
>
>  [1] Invisible Image Watermarks Are Provably Removable Using Generative AI
>
> ---
>
> Thank you once again for your valuable time. Your suggestions have improved our work, and we will incorporate the clarifications and additional experiments in the final version to address these concerns.

---

> > ### Author Rebuttal · Reviewer_gB3c · 2026-04-03
> >
> > I keep my point.

---

### Decision · Program_Chairs · 2026-04-30

**Decision:**

Accept (regular)

**Comment:**

This paper received mostly positive scores (4, 5, 4, 2). The proposed purification method is novel and interesting, different from existing methods that simply treat protective perturbations as random noise. The authors have put large efforts in addressing the reviewers' concerns, with extensive additional experiments and clear explanations. The only major concern, left by Reviewer 4, is about the proof in Section 3.2. In their communication with the AC, the reviewer states that "if the authors want to avoid this fundamental flaw, they should either remove the so-called proof in Section 3.2 and replace it with empirical analysis, or provide a real proof that adversarial samples indeed have a larger Jacobian spectral norm." The AC believes that, considering the empirical contributions of this paper, this issue is insufficient to ground a rejection, because a good paper does not necessarily provide a theoretical contribution.

One question for the authors: the current version does not contain any quality evaluation of the purified input images and related visualizations. To the AC's understanding, the attackers may not only care about the output but also the input. The AC suggests the authors comment on this point in the final version.